# ALCC: Migrating Congestion Control to the Application Layer in Cellular Networks

YASIR ZAKI●
*New York University Abu Dhabi*
*yasir.zaki@nyu.edu*

ROHAIL ASIM
*New York University Abu Dhabi*
*ra3106@nyu.edu*

MUHAMMAD KHAN
*New York University Abu Dhabi*
*mk7406@nyu.edu*

SHIVA IYER●
*New York University*
*shiva.iyer@nyu.edu*

TALAL AHMAD
*Google*
*mailtalalahmad@gmail.com*

THOMAS PÖTSCH
*New York University Abu Dhabi*
*thomas.poetsch@nyu.edu*

LAKSHMI SUBRAMANIAN
*New York University*
*lakshmi@nyu.edu*

## Abstract

TCP is known to perform poorly in cellular network environments. Yet, most mobile applications are explicitly built on the conventional TCP stack or implicitly leverage TCP tunnels to various cellular middleboxes, including performance-enhancing proxies, application-specific edge proxies, VPN proxies and NAT boxes. Despite significant advances in the design of new congestion control (CC) protocols for cellular networks, deploying these protocols without bypassing the underlying TCP tunnels has remained a challenging proposition. This paper proposes the design of a new *Application Layer Congestion Control (ALCC)* framework that allows any new CC protocol to be implemented easily at the application layer, within or above an application-layer protocol that sits atop a legacy TCP stack. It drives it to deliver approximately the same as the native performance. The ALCC socket sits on top of a traditional TCP socket. Still, it can leverage the large congestion windows opened by TCP connections to carefully execute an application-level CC within the window bounds of the underlying TCP connection. This paper demonstrates how ALCC can be applied to three well-known cellular CC protocols: Verus, Copa, and Sprout. For these protocols, ALCC can achieve comparable throughput and delay characteristics (within 3-10%) as the native protocols at the application layer across different networks and traffic conditions. ALCC allows a server-side implementation of these protocols with no client modifications and with zero bytes overhead. The ALCC framework can be easily integrated with off-the-shelf applications such as file transfers and video streaming.

## 1 Introduction

According to Cisco's recent Global Mobile Data Forecast Update [13], approximately 50% of the global data traffic is generated by mobile devices. Mobile applications primarily rely on TCP as the basic transport protocol, and TCP is known to perform poorly in cellular networks [14, 50, 51]. Packet losses are frequent events over cellular networks due to varying link rates, fading, burst scheduling, and unpredictable user mobility. TCP congestion control responds poorly to such losses, unnecessarily reducing the sending rate by half, thus sacrificing valuable network bandwidth. To cope with the above issue, cellular network operators often significantly over-provision their network buffers (at least

10x the bandwidth-delay product) and rely on lower layer retransmissions to shield the end-to-end TCP connection from these losses. But this is known to result in *bufferbloat* [17], a phenomenon in which end-to-end packet delays are very high due to the large buffers being kept full most of the time. Over the past decade, several new congestion control protocols have been proposed to replace TCP [4, 14, 50, 51]. Despite their superior performance over TCP, none of these solutions have gained wide deployment over real network conditions. Some factors limit the adoption of these new solutions:

- Cellular networks adopt specific packet-filtering and packet-shaping middleboxes that often limit the type of protocol traffic allowed in the network (e.g. UDP packets may be blocked), thus forcing algorithms to use TCP.

- Most mobile applications are built using advanced API calls, which explicitly establish HTTPS tunnels to specific cloud services or secure Remote Procedure Call (RPC) to middleboxes. Both of these use the standardized TLS/SSL layer, tightly coupled with the underlying TCP layer [43], which again makes it difficult to circumvent TCP.

- New congestion control protocols need to be ideally enabled/installed during Operating System (OS) upgrades, which are infrequent due to slow-release cycles and mainly due to their potential interruption in existing settings.

Elaborating on the first two points, there are many different scenarios that illustrate the fact that an underlying TCP connection at the edge is unavoidable: *i)* server-client TCP connection that goes through a NAT proxy in the cellular network; *ii)* a secure server-client connection that traverses an active middlebox that performs header inspection and HTTPS-specific operations; *iii)* a server-client connection that traverses a middlebox that splits TCP connections [18, 42]; *iv)* a mobile transfer service using a cloud service API where the traffic between the client and server is explicitly routed via the cloud service; and, *v)* communication using non-standard socket interfaces to transmit/receive packets. In all these scenarios, there is an underlying TCP connection between the mobile client and a middlebox at the edge of the cellular network. These connections suffer from the bufferbloat problem due to the excessive buffer provisioning by the cellular providers. TCP tunnels are very commonly used for: *(i)*

mobile devices that are behind a NAT/Proxy [9]; *(ii)* Evolved Packet Core (3GPP) for supporting cellular networking functions; *(iii)* Third-party applications/services built for using secure tunnels to middleboxes. Mobile applications and services have no control and visibility over the underlying TCP tunnels and are exposed to the standard send/recv interface as in a TCP connection. This illustrates an ossification of TCP in the context of mobile communications.

Another problem that plagues existing solutions to congestion control is that of implementation. Most solutions require kernel modifications and re-implementations depending on the underlying datapath [35]. The existing brittle development ecosystem that is dependent on TCP has forced new congestion control protocols to be primarily built on top of UDP, such as QUIC [20]. However, any application requiring reliability or security support will need substantial code changes to integrate with these protocol stacks. Moreover, QUIC is not designed to address the bufferbloat problem in mobile environments and also suffers from packet reordering issues in cellular networks [41].

And finally, existing solutions do not provide options to implement CC algorithms quickly over application layer protocols such as HTTP and HTTPS. Again, due to the widespread deployment of such protocols over TCP, any framework that provides this capability would enhance the ease of implementation of new CC algorithms. Thus, this paper addresses the following question: *Can we provide a framework for deploying new congestion control algorithms easily in mobile devices at the user-space that leverage the broad deployment of TCP and operate on top of the TCP and application-layer protocol stacks while offering similar performances as their native kernel implementations?* Such a framework would be able to facilitate the rapid innovation, deployability and evolution of new protocols for mobile applications at the application layer and above, without modifying any aspect of the conventional cellular network architecture.

To address this question, we present *Application-level Congestion Control (ALCC)*, a framework that executes CC protocols at the application-level to achieve similar performance as the native protocol while operating on top of an underlying TCP stack with multiple applications support. The ALCC framework enables careful packet pacing at the application layer regardless of the underlying TCP congestion window, thus limiting the traffic sent down to the TCP stack and thereby enforcing the sending rate of the application-layer congestion control protocol. Most TCP variants tend to ramp up to larger windows while significantly driving up the packet delays in cellular networks. ALCC constrains the application-level sending window to reduce bufferbloat in the system and maintains a low end-to-end packet delay. In the TCP tunneling context, ALCC specifically addresses the cross-congestion control interaction between the user-level CC and the underlying CC of the TCP tunnel in two ways: (a) When the underlying transmission layer socket issues a

blocking signal, ALCC uses this signal to control the transmission rate; (b) The blocking signal also affects the behavior of the higher layer protocols to adapt to the varying signal.

ALCC solves many of the issues highlighted earlier. Since it makes use of existing TCP stacks, it is quickly deployable on a large scale. It provides a quick solution to implementing new or existing algorithms without significant effort. ALCC can also be implemented over application layer protocols on mobile devices, thus making deployment on mobile devices significantly easier than before. Another essential advantage of ALCC is that it can support specific protocols like Verus and Copa with server-side integration alone without any client-side modifications and zero-byte header. For protocols like Sprout that require receiver feedback integration, ALCC needs to incorporate client-side changes. ALCC provides customized APIs which indirectly expose the same TCP Berkeley socket APIs to the application layer—making it easy to integrate into existing applications. For recently developed CC protocols [4, 50, 51], we demonstrate how easily these protocols can be blended into the ALCC framework and maintain the application layer congestion window in contrast to the underlying TCP congestion window. We show that these protocols within the ALCC framework imitate the original protocols' performances and attributes while sustaining comparable throughput and packet delay characteristics (within $3 - 10\%$) irrespective of the underlying TCP flavors.

The paper makes the following key contributions:

- A framework to implement CC protocols within or above the application layer that sits on top of the legacy TCP stack. This facilitates rapid innovation, deployability, and the evolution of new protocols for mobile applications.

- Show how new CC protocols may be integrated easily into ALCC (with minimal code changes) and demonstrate that these protocols achieve the same performance as their native implementations through rigorous testing.

- Integrate the ALCC framework into existing off-the-shelf real-world applications such as the *Bftpd* FTP server.

- Show how ALCC can support server-side integration for specific protocols without the need to modify the applications' client implementation. It also does not add any additional overhead (zero-byte overhead).

- Show how ALCC can support both client and server-side integration for specific protocols completely independent of a kernel module to intercept packets at the kernel. Both client and server are modified to support their own Acknowledgement and sequence numbers.

- A light ALCC Android library to assist client-side integration of congestion control protocols for mobile applications. An Android App that supports the ALCC java library for uplink file transfers.

## 2   Why ALCC?

We motivate the need to create a configurable framework for implementing new CC algorithms in the userspace on top of the conventional TCP substrate exposed by cellular networks.

**Middleboxes, Tunneling, and API Gateways:** Cellular network operators have deployed various types of middleboxes to make efficient use of their network resources and provide end-to-middle-to-end security for potential threats. This ubiquity has led to middle-box ossification, making them key control points in the cellular network architecture. Many of these middleboxes explicitly break the end-to-end connectivity model and support split TCP connections implicitly or explicitly [18]. As a result, many applications that run on cellular networks are implicitly tied to an underlying TCP connection, which runs on legacy TCP software, which is hard to change easily.

Recent years have seen a massive expansion in tunneling protocols, enabling the creation of Virtual Private Networks (VPNs), providing the illusion of a physical network to the user. Many users worldwide resort to mobile VPN clients to bypass censorship or access geo-blocked content, and more commonly, for privacy and security reasons. Even though VPNs encapsulate messages to traverse middleboxes, the encapsulation tunnel is terminated at the VPN server. The end-to-end CC is disrupted, and transport over encrypted tunnels may not allow other network entities to participate in CC. Finally, many mobile applications leverage API gateways that rely on HTTP variants and AMQP-like interfaces [2], essentially relying on an underlying TCP substrate to a gateway node. Netflix [32] and Amazon [1] are well-known public services that have adopted such API gateways.

**Large delays in cellular networks:** Mobile applications over TCP are known to experience considerable delays due to the complexity of the underlying cellular architecture. Recent cellular architectures [6] are known to employ large buffers to protect against packet losses.

This issue is well studied in prior work [10, 50, 51], which demonstrates that all known variants of TCP suffer from extensive delays in cellular networks since they aggressively set a large congestion window leading to excessive buffering at the base station and the gateway nodes. This is a crucial element of the motivation for designing a framework such as ALCC since ALCC is designed to *throttle the sending rate in the underlying TCP connection* thereby significantly reducing bufferbloat and packet delays and improving overall performance.

### 2.1   ALCC vs. popular related frameworks

Since the in-kernel implementations of congestion control protocols is a challenging task, especially at scale, many new congestion control protocols require useful libraries not supported by the kernel (i.e., libboost and alglib in Verus [51],

or Bayesian forecasts in Sprout [50]). These rely on implementing these protocols within the userspace over UDP instead. UDP, however, lacks the required security support that is needed by many applications. Besides, many firewalls and middleboxes are configured to drop UDP traffic due to the lack of congestion control or explicit connection setup/tear-down. This hinders the deployment aspects of the implemented UDP-based congestion control protocols. In contrast, TCP does not suffer from this problem, making it the perfect candidate for deployment. Hence, a possible solution that could combine the deployment benefits of TCP while allowing better congestion control logic is highly desirable.

ALCC leverages the kernel's TCP implementation while allowing developers to implement their congestion control logic within the userspace, effectively controlling the data flow down to the TCP stack and enforcing the application-level congestion logic to dominate TCP's default congestion control. ALCC shares similar goals as other popular frameworks in literature today, such as Congestion Control Plane (CCP) and Google's QUIC. Next, we will discuss the main differences between ALCC and these frameworks. Table 1 shows high-level comparisons to QUIC and CCP.

#### 2.1.1   ALCC vs. CCP

ALCC, in spirit, shares some of the design goals of CCP [35], in the sense of providing developers with a way to easily implement congestion control protocols within the userspace, thus enhancing the pace of development and ease of maintenance of congestion control algorithms. However, ALCC addresses a fundamentally different problem relevant to cellular networks: the middleboxes may use TCP tunnels and split TCP connections for performance reasons. In these scenarios, by throttling the traffic through the TCP connections, ALCC can reap significant benefits in performance. Unlike CCP, ALCC leverages the kernel TCP implementation without directly modifying the datapath. This allows ALCC to benefit from the wide deployment popularity of TCP.

ALCC has two implementations: 1) Kernel-based: that shares flow-level information (such as end-to-end delay, bytes in flight, etc.), as well as information from each TCP Ack (such as sequence number, bytes Acked, etc.) with the ALCC userspace program so that the CC algorithm can make use of them. 2) Non-kernel-based: both the client and server rely on their own sequence and acknowledgment numbers. However, this implementation requires modifying the client to send acknowledgments back to the server upon receiving the sequence number set in the packet header by the server.

The CCP framework implements CC algorithms in two pieces: i) datapath logic and ii) the actual CC logic. The datapath logic is a small piece of code written in a LISP-like syntax that exposes the kernel datapath variables required to be reported to the CC algorithm at what temporal granularity. On the other hand, the actual CC logic to control

Table 1: High-level comparison of QUIC, CCP and ALCC frameworks

| Features | Frameworks | | |
| --- | --- | --- | --- |
| | QUIC | CCP | ALCC |
| Congestion Control | Userspace implementation | Userspace implementation | Userspace implementation |
| Supported Transport | UDP | Any data path including UDP & TCP | TCP only |
| Reliability | Externally configurable | Externally configurable | TLS/SSL + TCP dependable (not configurable) |
| End Hosts Support | Mandatory Client & Server implementation | Supports Client only, Server only, or both implementations | Supports Server only or both Client and Server implementations |
| Re-transmission Mechanism | QUIC's packet number with directly encoded transmission order | - Based on TCP sequence numbers (ACKs) - Re-configurable | Based on TCP sequence numbers (ACKs). |
| Design | - Crypto handshake to minimize setup RTT - Pluggable Congestion Control Interface | Control Plane agent in the userspace enforcing rate and congestion window decisions via datapath modification. | Dominant userspace congestion control loop running atop TCP. |

the sending window is implemented in Rust or Python. In contrast, ALCC maintains a straightforward framework that enables CC algorithms to be easily ported by replacing native socket function calls (such as `send()` and `recv()`) with calls to the corresponding functions exposed by the ALCC library implemented in C++

### 2.1.2 ALCC vs. QUIC

QUIC's primary design goal focuses on "speeding up the web" by enabling multiplexed, encrypted, connection-oriented and reliable transport over UDP. However, the latest IETF Internet-Draft about the "Applicability of the QUIC Transport Protocol" necessitates QUIC's fallback to TCP [30]. It states that somewhere between 3% and 5% of networks block all UDP traffic. Therefore, all applications running on top of QUIC must either be prepared to accept connectivity failure on such networks or be engineered to fall back to some other transport protocol. In the case of HTTP, this fallback is TLS over TCP. QUIC has undoubtedly earned increased adoption and is currently the foundation for emerging protocols, e.g., HTTP3. An increasing number of distinct QUIC implementations exist today [21], including the mvfst framework from Facebook [26] (based on IETF QUIC's draft 29), with varying design goals, covering multiple programming languages. However, QUIC has not yet reached a stable RFC. From our observations in multiple discussion forums from developers trying to implement different congestion control algorithms over QUIC, it is a rather complex and not-so-straightforward task due to how a particular QUIC framework is designed, making it very difficult to integrate new congestion control protocols.

QUIC encrypts most of its packet header to avoid protocol entrenchment. However, a few fields are left unencrypted to allow a receiver to look up the local connection state and decrypt incoming packets. The main challenge for QUIC is to traverse firewalls that fail to detect QUIC packets and end up dropping them. QUIC falls back to TCP during persistent connection failures. Recent advancements in QUIC have resulted in some firewalls allowing initial packets but then blocking subsequent packets [39]. The most fundamental issue with QUIC is turning UDP into a connection-based protocol. This issue is intensified by middleboxes and firewall NAT services that hinder the process.

In contrast, ALCC allows a straightforward set of primitives and function calls to integrate any congestion control protocol within the framework. In fact, in this paper, we have combined three different congestion control protocols within the framework in just a few days. Additionally, once a congestion control protocol is integrated within ALCC, it becomes part of the library where any application can choose the suitable CC protocol. ALCC also makes it simple for applications to be integrated by exposing the same TCP Berkeley socket API.

## 3 Application-Level Congestion Control

To address the deployment challenges of recently proposed cellular CC protocols, we offer Application-level Congestion Control (ALCC) as a framework to execute these protocols at the application layer without modifying the underlying transport layer. ALCC explores the question: *Can we derive the performance benefits of new CC algorithms by deploying them at the application layer on top of traditional TCP stack?*

The aim is to find a way for a new CC algorithm to interact with existing legacy TCP stacks in a controlled manner. The goal is to get the advantages of both the new CC algorithms and the widespread deployment of the legacy TCP-based underlying architecture. We achieve this by leveraging TCP's buffers and using them to mimic the CC algorithm's sending behavior in the application layer.

### 3.1 Flying under the TCP radar

Each of many recently proposed protocols such as Sprout [50], Copa [4], and Verus [51] claim that their CC responds more efficiently in cellular network environments than existing protocols. Yet, despite their superior performance in a cellular context, actually deploying these protocols at scale remains a challenging task.

We suggest migrating these new CC protocols to the application layer to operate over the widely used TCP stack (without any modifications to TCP). To better understand

how this is achieved, let us first explore the following hypothesis: *the effective congestion window can be actively controlled from the application layer (i.e., pacing application data lower than TCP's CWND).* It is well known that the widely used TCP flavors, such as Cubic and Reno, maintain unnecessary large CWNDs and cause high packet delays. We evaluate two experiments that aim to test the following hypothesis: *is the TCP CWND affected by the amount of data sent from the application layer? Does it still maintain a large CWND even if there is not enough data to exploit this large CWND fully?* If this hypothesis holds, then the application layer can efficiently send data without incurring high packet delays or causing congestion/packet loss. This is achieved by throttling the application data flow to keep it below TCP's CWND. The control done in the experiments is achieved by limiting the application data flow sent down to the transport layer (we have chosen the standard TCP Cubic [23]). Unless otherwise stated, the tests were conducted on an experimental Mininet testbed consisting of a fixed bandwidth link of 12 Mbit/s, a client, a server and a router. In the first experiment, we implemented a shim layer within the application layer that maintains a static congestion window set to the theoretically required window to saturate the network link. In the second experiment, we implemented the shim layer to randomly choose the congestion window every second. Both experiments were tested on Ubuntu 18.04.1 with kernel version 4.15.0. Figure 1 compares the congestion windows of these two simple application layer protocols with the underlying TCP Cubic window. The solid lines represent the application layer window, and the dashed lines are the TCP Cubic congestion window. The figure confirms our earlier hypothesis that the TCP Cubic window is unnecessarily high, even higher than the application layer window. We made similar observations for TCP Reno and Bic. The bottom figure highlights the fact that we can arbitrarily control the sending window within TCP Cubic's envelope.

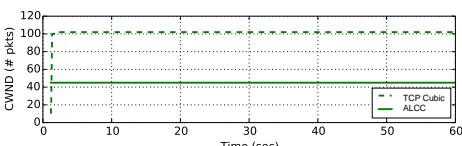

(a) Application layer control with static CWND

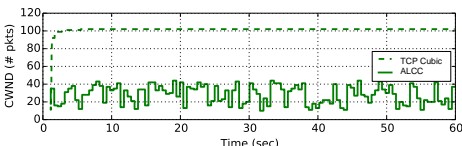

(b) Application layer control with random CWND

Figure 1: Application layer control and TCP Cubic CWND

Figure 2 shows how the application layer can control the

TCP Cubic congestion window to achieve better performance. Here, the application layer maintains a static congestion window and is compared against a legacy TCP Cubic network stack. We observe that the legacy TCP Cubic congestion window increases significantly over time, causing high packet delays. In contrast, controlling the window at the application layer (while still running on top of the TCP stack) achieves similar throughput without causing high packet delays. These results tentatively prove the hypothesis we discussed earlier by highlighting that one can perform a second CC loop within the application layer while leveraging TCP's limitation of maintaining a large congestion window. This motivates the idea of running CC protocols within the application layer without having to replace the TCP stack. An interesting observation in Figures 1 and 2 is the static behavior of TCP cubic CWND. This phenomenon is explained in Section 5.7.

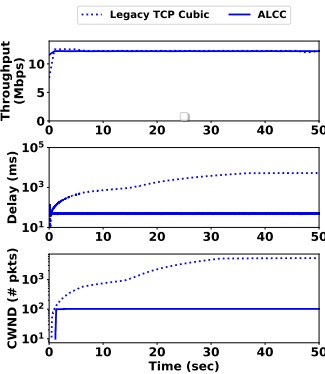

Figure 2: Application layer control with static CWND vs. legacy TCP Cubic throughput, delay and CWND.

## 3.2 Overview of ALCC

The core idea of ALCC is to perform CC at the application layer by *staying under the TCP radar*. ALCC's primary function is to replace the congestion window of TCP with the congestion window computed at the application layer by a new CC protocol. Based on this calculated window, ALCC tightly constrains the application data flow to the transport layer; the congestion window of ALCC is the *minimum* of the window size computed by the new CC protocol and the one of the underlying TCP protocol.

In essence, the ALCC framework emulates the cellular CC protocol at the application layer and computes the transmit window of the protocol. ALCC relies on the TCP socket interface feedback to implicitly learn the transmit window of the underlying TCP variants' protocol. For every packet transmission, if the TCP socket reports a full buffer and blocks on a potential transfer (or indicates full buffer in a non-blocking socket), ALCC delays the next packet transmission. More precisely, when the ALCC send function tries to send Bytes to the TCP socket, it will get a block signal in the form of an

error message (given the non-blocking flag set for the TCP socket). This stops ALCC from sending data down the TCP socket until the socket becomes available. Meanwhile, ALCC can still receive data from the Application if there is room in the ALCC buffer. However, if the ALCC buffer is also full, ALCC sends a block signal to the Application to stop sending new data. This blocking signal serves as an indication from the underlying TCP stack that the network might be congested and as a result the ALCC protocol needs to slow down.

The basic version of ALCC maintains a separate layer of acknowledgments at the application layer to keep track of the number of outstanding packets at the application level. This enables the ALCC layer to determine the packet transmission rate at the application level. Similar to Verus and Copa, ALCC can also support a server-side implementation where only the server can execute the CC protocol, and the client runs a native TCP connection with no application layer CC. In this case, the server needs to rely on TCP layer acknowledgments to track the outstanding window of packets.

The simple design of ALCC allows it to transmit at a lower rate than the TCP window and correspondingly achieve better delay characteristics. Surprisingly this strategy enables ALCC to achieve similar delay-throughput trade-offs as the native CC protocols while maintaining the fairness properties of TCP. By controlling the sending window, ALCC can reduce network buffer sizes and hence end-to-end delays without sacrificing throughput in comparison to standard TCP.

## 4   Realizing ALCC

The ALCC framework is implemented as a C++ library that acts as a shim layer connecting the application layer and the transport layer[1]. In other words, the ALCC C++ library is implemented in the userspace as a wrapper around the default Linux Berkeley sockets TCP implementation. It is designed to expose the same socket API to the application layer. This is performed to facilitate a smooth and easy integration of the ALCC library into existing applications. We describe three different implementations:

- Server-side ALCC library: This is the default ALCC implementation, which relies on the native TCP acknowledgments instead of the implemented CC protocol's acknowledgments. Here, the client is kept unmodified. This is realized by an ALCC kernel component that is implemented as a Linux kernel module. The ALCC kernel module is implemented with Netfilter and NetLink, and it acts as a cross-layer module to filter TCP acknowledgments and send them to the userspace program.

- Client/Server ALCC library: This is a special implementation extended from the above, without the use of the

kernel module. It relies on the implemented CC protocol's acknowledgments, and the client code is also required to use the library to send back acknowledgments. This implementation is meant for protocols that rely on external signals apart from acknowledgments, where some additional data needs to be shared by the client to the sending process.

- Mobile Java ALCC library: This implementation is meant for Android mobile phones that allows them to use ALCC to send data efficiently in the uplink direction. This implementation is similar to the Client/Server ALCC library, where it does not rely on a kernel module but rather implements its own acknowledgment mechanism.

A significant benefit of the default Server-side ALCC library is its single-side (server only) modification, which does not require any client changes. This is an exceptional advantage because it simplifies the deployment significantly, where ALCC relies on the underlying TCP stack for packet sequence numbers and acknowledgments. This is perfect for supporting the implementation of CC protocols such as Copa and Verus in the downlink direction, where TCP's acknowledgments would suffice. On the other hand, CC protocols such as Sprout do require additional information to be sent back to the sender from the receiver in addition to the acknowledgments. This is why we implemented the second library to deal with integrating these protocols. For example, Sprout's receiver sends back the observed packet arrival times as the primary signal to determine the network condition.

### 4.1   Server-side ALCC library

#### 4.1.1   ALCC Userspace Module

The userspace module is where the core part of ALCC is implemented; it is responsible for executing the application-layer CC protocols. We will call the ALCC userspace module as the *"ALCC library"*. The application uses the ALCC library to open an ALCC socket instead of a TCP socket. The ALCC framework of the library is shown in Figure 3a. The main philosophy of the ALCC library is to provide placeholder functions to integrate any CC implementation easily. The idea is to split the CC implementation into three processes: i) basic CC logic, ii) sending-related functionality, and iii) receiving-related functionality. The sending mechanism of ALCC also opens a standard TCP socket to send data down to the transport layer. The ALCC library is implemented as a C++ class, where the core part of the implementation consists of a:

1. Circular queue implementation

2. Sending thread (`ALCCSocket::pkt_sender`)

3. Receiver thread (`ALCCSocket::ack_receiver`)

4. CC logic thread (`ALCCSocket::CC_logic()`)

The circular queue is used to store the data sent down from the application layer so that ALCC can pace the MTU-sized packet sending based on the CC sending mechanisms. Introducing an intermediate queue allows ALCC to leave the sending mechanism of the application unchanged. Like the standard TCP `send` buffer, the ALCC framework blocks the application once the intermediate queue is full, which may occur if the underlying TCP sending kernel buffer is full.

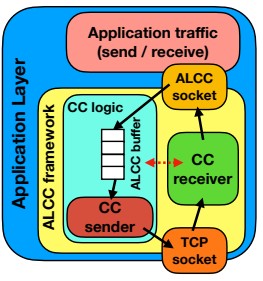

(a) User-space module

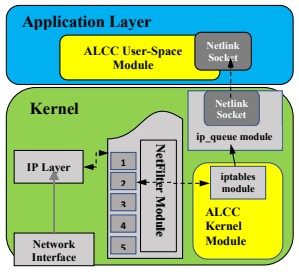

(b) An example of packet filtering used in Linux kernel

Figure 3: ALCC library framework architecture.

The sending thread is responsible for sending data packets from the ALCC queue down to the TCP stack using a regular TCP socket. It paces the data sent down by relying on the CC logic thread's decisions to regularly determine the so-called "application layer congestion window" (analogous to the transport layer congestion window). The application layer congestion window is computed by the CC protocol running within the ALCC library. The receiver thread implements the CC protocol logic when receiving an acknowledgment. As stated earlier, ALCC does not implement its own acknowledgment but rather relies on the underlying TCP ACKs. These ACKs are sent by the ALCC kernel module running within the Linux kernel. For the application layer, ALCC exposes two function calls: `alcc_accept()` and `alcc_send()` that are meant to replace TCP's `accept()` and `send()` functions. These are the only modifications required at the application to use ALCC. The new function calls are intentionally kept similar to the default TCP ones—in terms of the arguments they use—to ease the integration as:

```
int alcc_accept(int sockfd, struct sockaddr *addr,
    socklen_t *addrlen)
int alcc_send(int sockfd, const char * buffer, int
    buflen, int flags)
```

Alternatively, we can also simply use the `LD_PRELOAD` [19] trick to change the TCP socket system calls `accept()` and `send()` to the ALCC calls, which makes ALCC even more usable since there won't be a need to replace the system calls within the applications.

Within the `alcc_accept()` function, the basic TCP `accept` function call is performed, and the corresponding

TCP socket is passed to the ALCC framework object instantiation. This socket is then used by the ALCC object to send and receive data. As for the `alcc_send()` function call, it mainly accepts the data from the application and then inserts it to the ALCC internal buffer to be sent later by the ALCC sender thread.

### 4.1.2  ALCC Kernel Module

A couple of recent delay-based CC protocols, such as Verus and Copa, rely on their own acknowledgments to infer what is happening in the network. Their native implementations rely on the UDP protocol as the underlying transport protocol. With the introduction of the ALCC library, and because ALCC runs on top of TCP sockets, an opportunity arises to simplify the integration efforts in real-world applications. Some of the ultimate design goals behind any successful protocol are simplicity and efficiency. To achieve a zero-byte overhead and server-side only modifications, we modified the Verus and Copa implementation within ALCC to rely on TCP ACKs rather than on their own. In our implementation, the TCP ACKs are retrieved through an ALCC kernel module that we built using the Linux kernel framework known as Netfilter [5]. Netfilter offers packet manipulation via various hooks into the network layer. We have used the NF_IP_PRE_ROUTING hook, triggered by any incoming traffic soon after entering the network stack, and before the kernel performs any routing choices for packet sending. Figure 3b shows an example of multiple components fitting together to provide an insight into how filtering and communication between the kernel module and the userspace module are achieved. When an IP packet arrives at the network layer, the kernel sends the packet to the Netfilter module, which then transfers it to the iptables module. The latter holds a set of rules defined by the ALCC kernel module to specify the actions to be taken when the desired packet is detected. It first inspects the transport layer type within the IP packet. In the case of TCP, it extracts the TCP header and checks the destination port number. If an ALCC socket has already registered that destination port, the hook function extracts the ACK details and sends it to the respective ALCC userspace process. For the kernel-to-userspace delivery, the ip_queue module uses Netlink sockets. We have implemented a signaling protocol between the ALCC userspace module and the kernel module. When a new application opens an ALCC socket, the ALCC socket first opens a legacy TCP socket, which gives back to the ALCC framework the actual port number used with this socket. The ALCC userspace module would then send a `port_registration` message to the kernel module to register its process id with the associated port number. The ALCC kernel-module maintains a mapping table between the different port numbers and their corresponding ALCC userspace process IDs. When the ALCC flow is finished, it sends a `port_release` message to the kernel module to

remove the registered port number. When CC protocol designers use the ALCC library to integrate/implement their protocol, there will be no requirement to implement any functionality within the ALCC kernel module.

### 4.1.3 CC protocols integration into ALCC

To evaluate the ALCC framework's performance, we ported two protocols into the ALCC library – Verus and Copa. In addition to the significant benefits of running a CC protocol within the application layer, the time and effort taken to integrate such a protocol are equally crucial.

**Verus integration:** The Verus integration was straightforward since its threads fit well in the structure of the ALCC library class. Verus has four main threads: a sending thread, a receiver thread, a logic thread, and a delay profile thread. When it came to porting the Verus code base into ALCC, one of the main changes was integrating the receiver thread and the Verus acknowledgments. Verus uses a header that consists of a sequence number, CWND when the packet was sent and the sent time. Since the main target of ALCC was to simplify the integration effort and keep it bound only to the server-side, we relied on the underlying TCP's sequence numbers and acknowledgments instead of using Verus's own sequence numbers and acknowledgments. We also created two different mappings within the ALCC library to store both the congestion window at the time the packet was sent and the sent time (so as not to carry these fields within the Verus header, thus allowing us to remove the Verus header all together).

Two main challenges in relying on TCP's ACKs were obtaining the sequence numbers at the application layer (since TCP runs at the kernel), and TCP's sequence numbers are bytes-offset. In contrast, Verus's sequence numbers are simple integers. The first challenge was solved using the ALCC kernel module. The second challenge was solved by maintaining a mapping between TCP sequence numbers and the sequence numbers of Verus. During the initial phase, ALCC must listen for the first sequence number exchanged between the client and the server during the TCP handshake, since TCP chooses the starting sequence numbers at random. The Verus code base took about a day to port into ALCC.

**Copa integration:** We leveraged the generic CC implementation of Copa [3]. The main challenge was that Copa's code implements four different CC protocols: Copa, Remy, kernel CC (Cubic on Linux), UDT's [22] TCP AIMD implementation and PCC (deprecated). All of these protocols, including Copa, were implemented using UDT's class. The challenge was to extract Copa's codebase and any additional required code from the UDT class. Luckily, the main Copa's implementation was bounded to the `markoviancc.cc` and `markoviancc.hh` files. As for the main logic of the sending/receiving packets we extracted the code from the `ctcp.hh`, specifically `CTCP<T>::send_data` function. This function handles the sending of packets and then checks if any data is pending to be received. We had to split the function into two halves, where the sending code was moved into ALCC's sending thread, and the packets' acknowledgments handling logic was moved into the receiver thread. We had to duplicate some of the variables that were used by both parts. We faced some challenges within the receiver logic because TCP is a byte stream protocol where TCP sometimes acknowledges multiple packets in a cumulative acknowledgment. Since Copa's receiver logic was built to handle a single packet acknowledgment, we first figured out how many packets are being ACKed and then wrap Copa's receiver logic with a loop that can handle the multiple acknowledgments. Additionally, Copa uses a map for `unacknowledged_packets`, which before was protected since the sender and receiver logic were part of the same function and were executed one after the other. Due to the split in the ALCC framework, we had to protect this map from corruption by simultaneous access by both threads, which was achieved using mutex locks. Like Verus, we had to add mapping to store Copa's packet sent times to compute the packet delays. We also had to handle TCP's byte-offset sequence numbers and map them to Copa's integer-based sequence numbers. The porting of Copa's codebase took about two working days.

## 4.2 ALCC without a kernel module (Client/Server ALCC library)

This library implementation is very similar to the above implementation, except for the following key aspects. First, this library does not require the ALCC kernel module since it relies on acknowledgments sent by the client-side CC protocol implementation rather than the TCP ACKs. Second, the ALCC userspace module is almost identical to the server-side library implementation, except for how the acknowledgments are handled. Given that we no longer have access to the TCP Kernel, there was no way to rely on TCP's sequence numbers and acknowledgments. Instead, the ALCC userspace sender module relies on acknowledgments sent by the client to the application. In the client-side modification, the ALCC library modifies the TCP `receive` function call to first read the CC protocol header from within the TCP socket and then send back an ACK as a separate packet to the server. The CC protocol packet payload is then read and returned to the client app as valid application data. Of course, here, due to the byte-stream nature of TCP, the CC protocol payload length is determined from the packet header. The above ACK mechanism is simply an application layer acknowledgment implementation. Where each ALCC packet will have its own sequence number, and ALCC would record the exact time when that packet (i.e., sequence number) was sent into the network. Then at the client-side, for each received ALCC packet, the client would send back an ACK with the corresponding sequence number. Upon receiving such an ACK at the ALCC sender, ALCC can calculate the round-trip time by

taking the difference between the receive and the stored send times. This RTT is then internally used by the CC algorithms, whether Verus or Copa, to update the sending rate. Next, we highlight the noticeable pros and cons of ALCC with and without the kernel module as follows:

- ALCC with kernel module requires only server-side modification, whereas server and client-side changes are required in the non-kernel ALCC.

- ALCC non-kernel version significantly reduces the CPU utilization. In contrast, ALCC with kernel module increases the CPU utilization by two folds compared to the utilization of native CC protocol.

- Adding a kernel module to mobile phones can be challenging due to constraints imposed by the mobile operating systems. Therefore a non-kernel implementation would be a more straightforward alternative for ALCC implementation in mobile phones.

### 4.3 Implementing ALCC on a mobile device

Mobile operating systems, such as Android, impose restrictions on modifying the underlying TCP stack, not providing access to kernel modules. To integrate ALCC into a mobile OS, we have developed a preliminary Java library that supports running ALCC atop application layer protocols to circumvent the aforementioned restrictions. This approach has multiple advantages, such as maintaining the TLS/SSL connections and keeping the underlying TCP implementation untouched or transparent to (reverse) proxies that might intercept the connection along the way.

The main task of the library is to send the chunked application data packets as separate TCP packets/frames and receive acknowledgments from the server to obtain the round-trip time estimates. Unlike the desktop Linux implementation, which uses a kernel module for cross-layer communication, we implemented our own acknowledgment mechanism in this Android library by adding a custom header with sequence numbers, thus obviating the need for a kernel module. We also extended the ALCC server library to send back ALCC acknowledgments with the sequence numbers. This library would be handy for mobile applications such as video conferencing, live streaming, or social media apps.

Integrating an existing version of a CC protocol into the Android ALCC library can be challenging because many of these protocols leverage several external libraries currently not supported by the Android OS. Ideally, alternative libraries can be utilized or even implemented to overcome the missing/unsupported ones. However, this can be an exhaustive and time-consuming task, which is out of scope for this paper. Thus, to be able to quickly provide a proof-of-concept implementation of one of the new cellular CC protocols into the ALCC Android library, we relied on using the Model-Driven Interpretable (MDI) congestion control [29] approach. MDI

allows approximating any congestion control algorithm as a general discrete-time Markov model by a 2-dimensional state space, represented in the form of a state-transition probability matrix for that algorithm. Each state is a tuple of the relative change in the network delay and the sending window size. The matrix describes transition probabilities between every pair of states and is obtained by training the algorithm on a large set of network configurations. MDI versions of popular algorithms mimic the actual throughput and delay characteristics of algorithms on real traces. Thus, using the publicly available Markov models trained independently over diverse network conditions, the behavior of many algorithms atop ALCC on Android.[2] can easily and effectively be replicated.

We have built an Android App that utilizes the java Android ALCC library to upload files from the phone memory to the server. The App relies on MDI as the main congestion control logic. We have utilized a transition probability matrix of Verus that was trained over 1000 cellular traces as described in [29]. Figure 4 shows the throughput and delay performances of the Android ALCC app(using Verus MDI) compared to the TCP cubic performance measured using an upload of the same file with *scp*[3] from a laptop that was tethered to the Android phone. Several experiments were conducted to upload a large video file to the server using the App, and a laptop tethered to the Android phone over a real 3G network. It is observed that the throughput for both Android ALCC and the scp upload TCP are approximately alike when analyzing several experiment runs. However, the delays achieved using the Android ALCC library are much lower in comparison to the scp ALCC version. This also highlights the foremost advantage of the ALCC framework when testing congestion control algorithms for uplink in cellular networks.

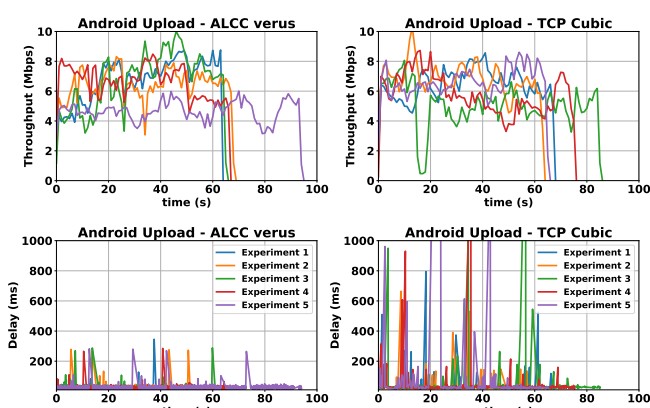

Figure 4: ALCC (with MDI on Android) vs TCP cubic (on Android): Throughput and Delay performances over real 4G cellular network.

---

[2] https://github.com/comnetsAD/MDI

[3] *scp* is linux program that is used to copy files between a local and a remote system or between two remote systems.

## 4.4 Integration into off-the-shelf Applications

In this section, we discuss and highlight the integration efforts of the ALCC library into existing off-the-shelf applications. To demonstrate that the ALCC library can easily be integrated into real applications, we chose a number of off-the-shelf applications as a proof-of-concept examples for the integration: FTP using the Bftpd server [45], web using the Ryuuk concurrent web server [34], and video streaming using the RTMPServer for Adobe Flash player [31].

First, we started integrating ALCC into the bftpd server application. Since easy integration was one of the main motivations behind designing the ALCC framework, we exposed three different ALCC functions to act as a replacement for their TCP counterpart within the application implementation, these are: `alcc_accept()`, `alcc_send()`, and `alcc_close()`. We searched within the bftpd application for the location where they instantiate the TCP socket, mainly looking for the TCP `accept()` call, which we found inside the `command.c` file. We then simply replaced the `accept()` call with `alcc_accept()`. The latter is implemented to instantiate the ALCC object, which would create the ALCC queue and the multiple ALCC threads. We then replaced the `send()` function call with our own `alcc_send()`. However, due to the byte-stream nature of TCP, we had to enclose the sending function in a `while` loop that can guarantee the complete sending of all the required bytes. Finally, we had to replace the TCP `close()` call with the `alcc_close()` call so that we can make sure that all data stored within the ALCC queue are sent first before terminating the connection. For the RTMPServer and the Ryuuk web-server integration, we had to do the same as above. We replaced the above three TCP function calls with their ALCC counterparts. That is mainly found inside the `main.cc` file for the RTMPServer, and the `SocketListener.cpp` and `SocketStream.cpp` for the web-server.

## 5 Evaluation

Our evaluation demonstrates that the ALCC framework, with three different integrated CC protocols (Verus, Sprout, and Copa), can achieve the same throughput and delay distribution characteristics as their native protocols regardless of the underlying TCP transport protocol. The evaluation were conducted on an Ubuntu 18.04.1 machine with kernel version 4.15.0, with Intel Xeon(R) CPU E3-1246 v3 @3.50GHz x 8 with an 8GB RAM.

Applications that run on top of the ALCC stacks would naturally observe similar behavior as the native protocol under different network conditions. The main goal of our evaluation is to demonstrate how the integrated CC protocols within the ALCC framework closely match their native protocol performance including their temporal characteristics. To demonstrate reproducible results and control for different aspects

of the evaluation under the same network conditions, we collected a diverse set of cellular network traces and used the network emulation environment Mahimahi [36] that enabled us to test different protocol implementations across different network conditions in a controlled manner. Our ALCC implementation runs over real mobile networks, and we have conducted several tests running various off-the-shelf applications with ALCC over 4G networks.

## 5.1 Channel Traces

We compare the performances of Verus, Sprout and Copa over different TCP variants (Cubic, Bic, and Reno) within the ALCC framework. The experiments are conducted using the Mahimahi network emulator with various channel traces, some taken from published papers and others recorded from real cellular networks. **4G Verizon:** taken from [50] and represents a recorded channel over Verizon's 4G network in the US. **Rapidly changing network:** inspired by [14], this trace represents a highly fluctuating channel, where the magnitude is varied randomly every 5 seconds. **3G Etisalat:** taken from [51] and represents a recorded channel over the Etisalat 3G network while driving on a highway at 120 km/h.

Other channel traces are collected by setting up a server located at a University campus and four Android smartphones. A bi-directional setup was used to monitor the downlink and uplink channels using a 3G HSPA+ cellular network. Both the server and client concurrently send UDP packets of 1400 bytes. Data rates of 2.5 Mbps and 5 Mbps were set for uplink and downlink, respectively. However, these data rates do not necessarily indicate the maximum capacity of the cellular network. We assume that the channel is not over-saturated by using these data rates, and packet-buffering is minimized under perfect channel conditions. Measurements of three different scenarios are captured with varying properties of mobility. The scenarios are **City Drive, Campus Walk, Highway Drive, and Beachfront Walk**. The channel traces are generated from the packet arrival timestamps at the receiver and the inter-arrival times between consecutive packet arrivals. Additionally, the channel traces from all our four phones were combined into one large trace to emulate significant user contention.

## 5.2 ALCC-Verus vs. Verus

This section highlights the performance comparison of the ALCC-Verus implementation on top of legacy TCP Cubic stack and the native Verus. We examine the throughput and packet delay for both cases. The experiment is conducted using the Mahimahi emulator using multiple channel traces, with a bottleneck buffer size of 2 MB. We only show the results for four of the traces due to space limitation (City drive, 3G highway drive, Beachfront walk, and rapidly changing channel). Figure 5 shows the comparison of the achieved

instantaneous throughput and delay over time for the selected four channel traces, where the upper part of each sub-figure depicts the throughput, and the lower part of the figure highlights the delay performance in logarithmic scale. The results show that both protocols deliver almost identical throughput and packet delays on all channel traces. It can be observed that ALCC-Verus inherits Verus properties, such as avoiding network buffer overfilling, while fully saturating the link capacity. Worth mentioning here that the ALCC-Verus used in these experiments ran on top of TCP Cubic.

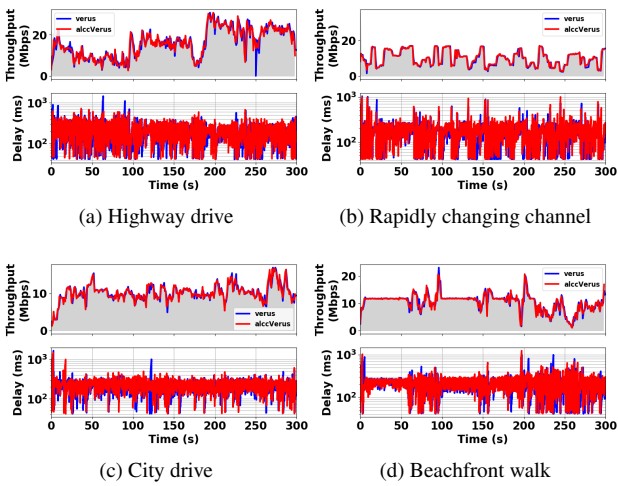

Figure 5: Instantaneous throughput and delay over time.

To accurately measure the performance similarity between the ALCC-Verus and Verus, we computed the Probability Density Function (PDF) for both the throughput and the delay using the Seaborn kernel density estimate [47]. These PDFs were calculated over 20 independent runs for each channel trace to obtain statistical significance. Figure 6 shows the PDFs for the selected channel traces, where the above part of each sub-figure shows the throughput PDF, whereas the lower part shows the delay PDF. From the comparison, it can be noticed that the ALCC-Verus PDF, depicted in red, does match the shape of the original Verus PDF shown in blue for all channel traces. Apart from some negligible marginal delays, variations are seen in the distribution of the rapidly changing network delay. Figure 7 shows an overall summary of the results comparing the different values of the results population. Each protocol is depicted by a circular shape representing the operational region of the protocol circumscribed by the 25% and 75% percentile of the obtained throughput and delay, where the crosses (x) indicate the median values. The lower and upper part of the shape represents the 25% and 75% of the throughput, respectively (y-axis) whereas the left and right part of the shape represents the 25% and 75% of the delay, respectively (x-axis). The results show that even though ALCC-Verus runs on top of TCP, it is still capable

of achieving similar statistical performance in terms of delay and throughput with a minor delay penalty not exceeding 15% in the worst case scenario (i.e., rapidly changing channel).

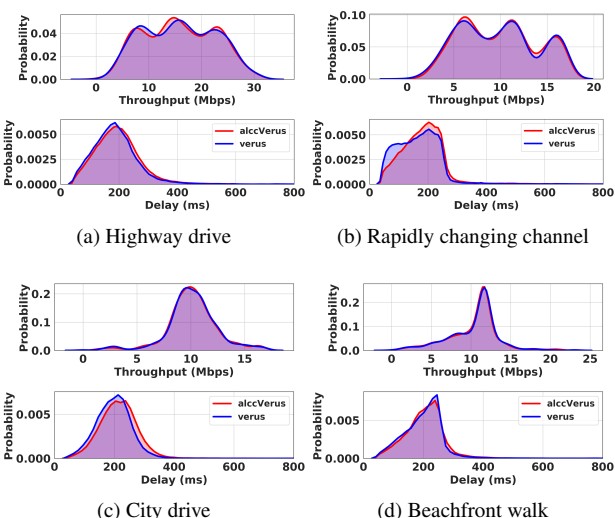

Figure 6: ALCC-Verus vs. Verus: Throughput/delay PDF

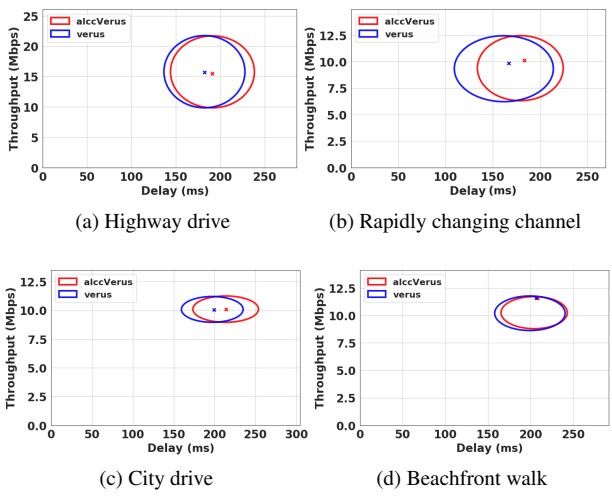

Figure 7: Summary of throughput and delay (population)

## 5.3 ALCC-Sprout vs. Sprout

This section highlights the performance of the Client/Server ALCC library implementation using Sprout as the use-case scenario. We evaluated ALCC-Sprout over multiple other channel traces, and it has shown similar results to the one discussed in this section. However, we have not presented them in the paper due to the page limitation.

Figure 8 shows the performance comparison of ALCC-Sprout vs. Sprout; this was performed using the rapidly

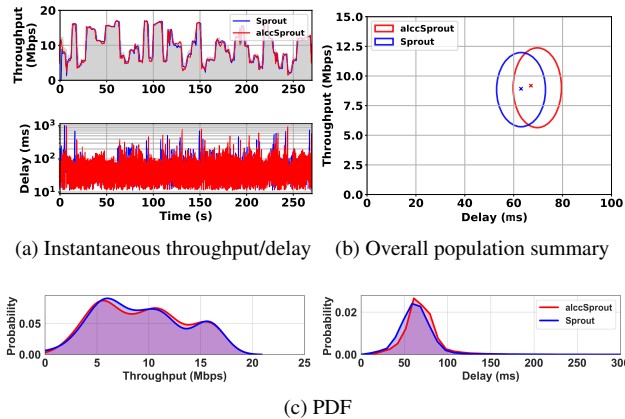

(a) Instantaneous throughput/delay (b) Overall population summary

(c) PDF

Figure 8: ALCC Sprout vs. Sprout: Throughput/delay (Rapidly changing channel)

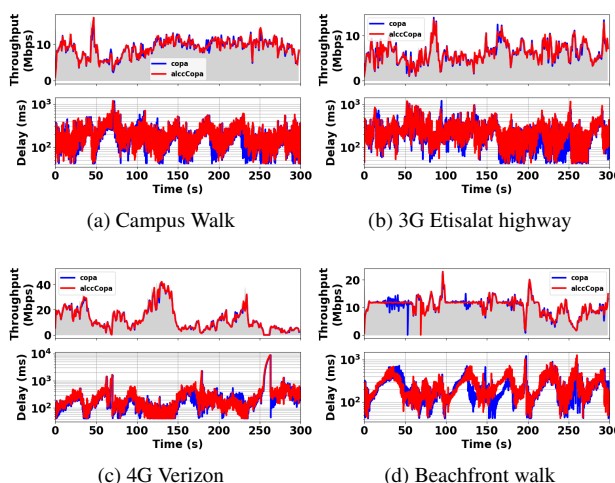

(a) Campus Walk (b) 3G Etisalat highway

(c) 4G Verizon (d) Beachfront walk

Figure 9: ALCC-Copa vs. Copa: Instantaneous throughput and delay over time.

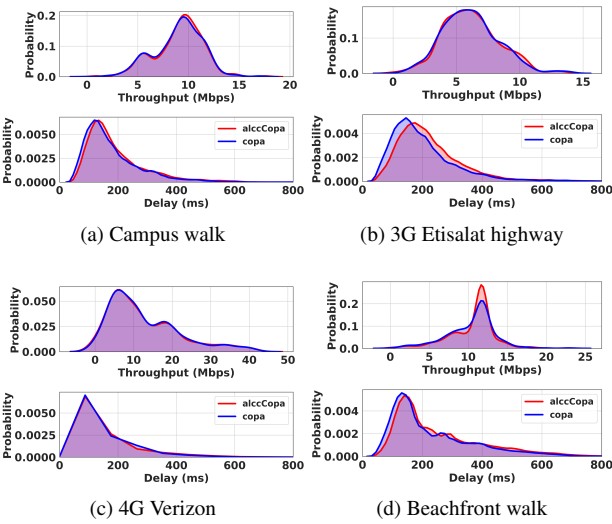

(a) Campus walk (b) 3G Etisalat highway

(c) 4G Verizon (d) Beachfront walk

Figure 10: ALCC-Copa vs. Copa: Throughput/delay PDF.

changing network trace. The instantaneous throughput and delay results, depicted by Figure 8a, demonstrate that ALCC-Sprout matches the original Sprout performance, thoroughly saturating the channel capacity and having similar delay characteristics. This can be further proven by the PDF results shown in Figure 8c, where the ALCC-Sprout PDF matches the distribution of the native Sprout protocol. Finally, the population summary results show that the ALCC-Sprout version incurs a slightly higher end-to-end delay than the original version for both the 25% and 75% of the population. However, the difference over the median is marginal.

## 5.4 ALCC-Copa vs. Copa

Figure 9 shows the throughput and delay comparison of native Copa versus ALCC-Copa (running on top of TCP Cubic). We chose to show the results achieved over the following channel traces: *4G Verizon, 3G highway drive, Campus walk, and beachfront walk*, performing 20 independent runs per trace with the same characteristics defined in the previous subsection. Although we show the results for these four channel traces, the other traces show similar performance and are omitted due to space restrictions. Similar to the ALCC-Verus results, it can be observed in Figure 9 that ALCC-Copa does achieve nearly equivalent instantaneous throughput and delay to Copa. The PDFs of the throughputs and delays of the two protocols are shown in Figure 10. It can be seen that ALCC-Copa achieves near-identical distributions to Copa. Figure 11 shows that the operational region of the protocols does match in all traces, despite some minimal difference in the highway traces. Moreover, from the median values, it is evident that ALCC-Copa achieves the same throughput, deriving all properties of the original Copa.

## 5.5 ALCC with Multiple TCP Flavors

In the previous sections, we have demonstrated that three different ALCC protocols achieve similar throughput and delay characteristics to their native protocols while operating on top of TCP Cubic. In this section, we investigate if the same holds if the ALCC protocols operate on top of other popular TCP variants such as TCP Bic, and TCP Reno. Figure 12 and 13 show the overall operating region results and the throughput and delay PDFs for both ALCC-Verus and ALCC-Copa, respectively. The results show a comparison to different underlying TCP flavors. These experiments were conducted over all the channel traces; however, we show the results of two traces per protocol. The results confirm that

ALCC does exhibit similar delay and throughput characteristics irrespective of the underlying TCP flavors.

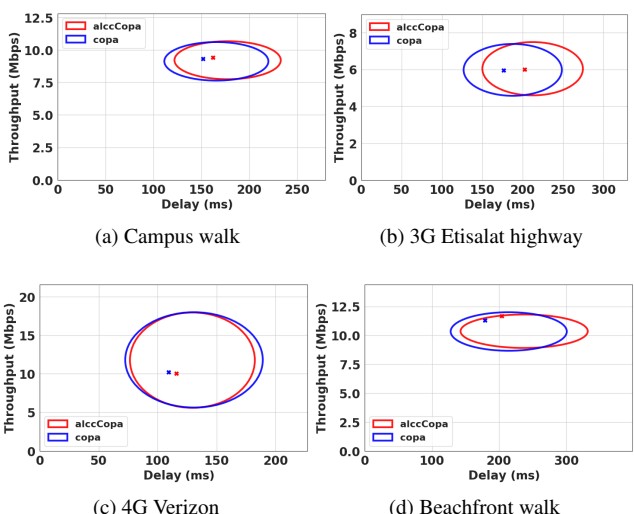

(a) Campus walk    (b) 3G Etisalat highway

(c) 4G Verizon    (d) Beachfront walk

Figure 11: Summary of throughput and delay (population).

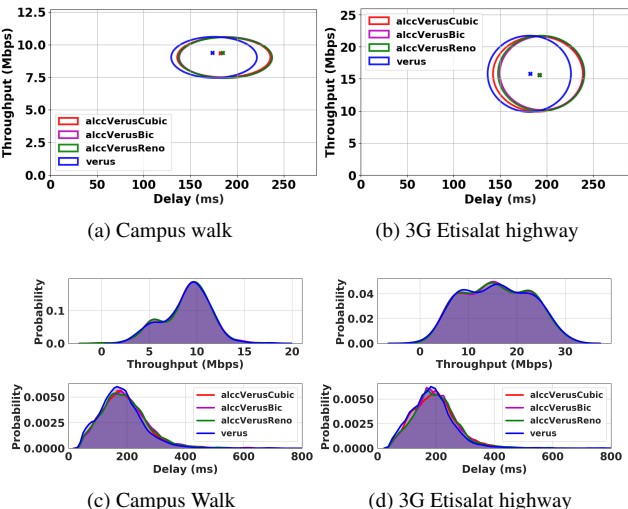

(a) Campus walk    (b) 3G Etisalat highway

(c) Campus Walk    (d) 3G Etisalat highway

Figure 12: ALCC Verus vs. Verus: Throughput/delay population and PDFs over different TCP flavors.

## 5.6 CPU utilization

Benchmarking the CPU utilization is essential for evaluating the ALCC framework compared to the native CC protocols. We demonstrate the userspace CPU overhead–caused by ALCC–by running both Copa and Verus with and without the ALCC framework. This evaluation is done over a cellular network environment, utilizing the 4G Verizon channel trace in the Mahimahi emulator. TCP Cubic is used as the

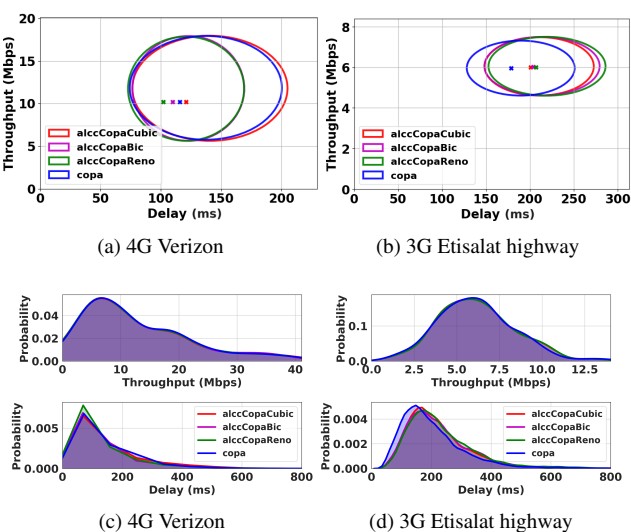

(a) 4G Verizon    (b) 3G Etisalat highway

(c) 4G Verizon    (d) 3G Etisalat highway

Figure 13: ALCC Copa vs. Copa: Throughput/delay population and PDFs over different TCP flavors.

underlying transport layer protocol. We measured the average userspace CPU utilization for Verus and Copa (with/without ALCC) running over a server with Intel Xeon E3-1246 v3 Octa-core (8 Core) 3.50 GHz Processor. The results have shown an increase in the CPU utilization for both Verus and Copa, ranging from 1.5x – 2x compared to the native versions without ALCC. Similar observations were also reported by the authors of CCP [35], which is mainly caused by the Inter-Process Communication (IPC) overhead. In ALCC, the cross-layer TCP ACKs are being sent from the kernel module to the userspace ALCC module. We have implemented another version of ALCC that does not rely on the Netfilter hooks to extract packet information at the kernel. Instead, the new implementation relies on its own Ack and sequence number mechanism. As such, this implementation requires a client-side modification to read and acknowledge the received packets appropriately at the ALCC level.

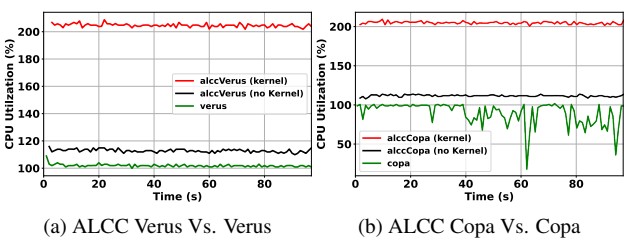

(a) ALCC Verus Vs. Verus    (b) ALCC Copa Vs. Copa

Figure 14: CPU utilization comparison over highwayGold channel trace.

As shown in Figure 14, the average increase in CPU utilization for ALCC with no kernel module is approximately 11% compared to the native version. In comparison, ALCC

with kernel module is almost 2x the CPU utilization compared to native Copa. This is because the current ALCC implementation is not optimized to reduce the Inter-Process Communication (IPC) and the cross-layer TCP Acks sent from the kernel; further optimizations can reduce the CPU overhead significantly.

For the Java Android ALCC library, the CPU utilization should be significantly lower than the server-side implementation. In this library, we do not rely on a kernel module to get the acknowledgment. Given that native Copa or Verus can not operate on the native Android mobile in the uplink direction, phone measuring the CPU overhead was not feasible.

## 5.7  ALCC in real cellular network

We further investigate how ALCC performs in a cellular network. We evaluated Verus as the primary congestion control in ALCC with Cubic as the underlying default TCP transport layer protocol. Figure 15 shows the logical diagram and the experiment setup. The client machine is tethered to an android phone via a USB connection. The Android Phone has 3G and 4G wireless support. The wireless setting is switched between 3G and 4G according to the experiments. A server is placed in another location that is physically connected through Ethernet to a home network with a fiber-to-home connection provided by the local ISP.

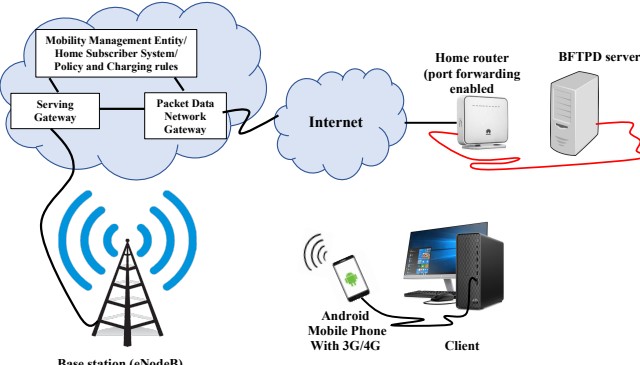

Figure 15: A real cellular network setup between the client and server for ALCC performance testing.

It is interesting to observe that in both cellular scenarios—3G and 4G— ALCC always operate with a smaller CWND than the underlying TCP CWND (green vs. red curve). The results also highlight that there are negligible packet losses observed by TCP due to the cellular networks' link-level retransmission and Automatic Repeat-reQuest (ARQ) recovery mechanisms. This allows the underlying TCP cubic to open a larger CWND, allowing ALCC to operate sleekly beneath it. Figure 16 and 17 show the CWND comparison of ALCC with TCP cubic as an underlying transport along with the native protocols. It is observed that the underlying TCP CWND

stays static after a specific time, which explains the negligible packet losses in the cellular networks due to the over-dimensioned buffers. However, TCP's congestion window is expected to grow persistently instead of remaining steady in the case of no packet loss. Although, [28] describes this phenomenon as a deliberate cap on the maximum advertised receive window (tcp_rmem_max) by smartphone vendors. We believe that this phenomenon is a result of the maximum size of the TCP send buffer. To confirm the above hypothesis, we studied the correlation between TCP's CWND and the TCP send buffer size (defined by the SO_SNDBUF). We performed multiple experiments using a file transfer scenario between an FTP server (bftpd) and a separate Linux FTP client. The server was configured to use TCP CUBIC and to systematically increase the TCP send buffer size in each experiment, mainly to 100KB, 500KB, 1MB, and 5MB. We monitored TCP's CWND in each case and confirmed that the maximum send buffer size of TCP directly correlates with the flat value of the TCP CWND as shown in Figure 18.

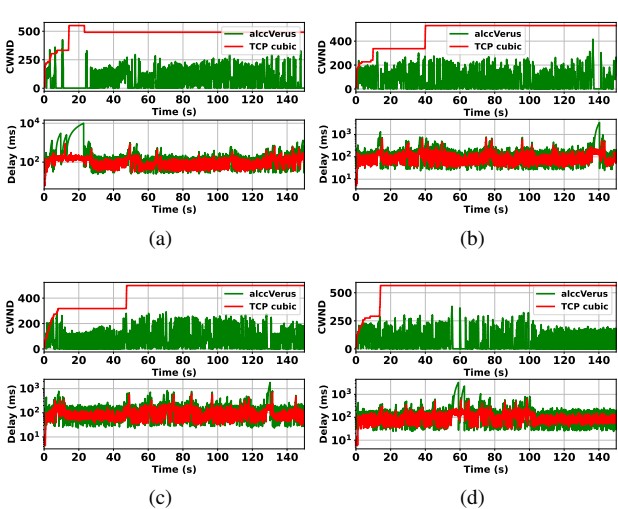

Figure 16: ALCC Verus and underlying TCP Cubic CWND analysis over a 3G network.

Figure 19 shows the throughput and delays performance of ALCC Verus compared to native Verus in a real 3G cellular setting, as shown in Figure 15. Although it is impossible to have the same channel conditions in real cellular setting evaluations; However, we tested ALCC Verus and native Verus for a stationary user over a 3G cellular network, relying on repeating the experiments multiple times to give us the same statistical fairness when comparing both native and ALCC protocols. We observe that both the throughput performances of ALCC Verus and native Verus are statistically similar, as shown in Figure 19. However, it can be observed that the delay performance of the native Verus protocol is slightly higher and occasionally suffers from multi-second delays.

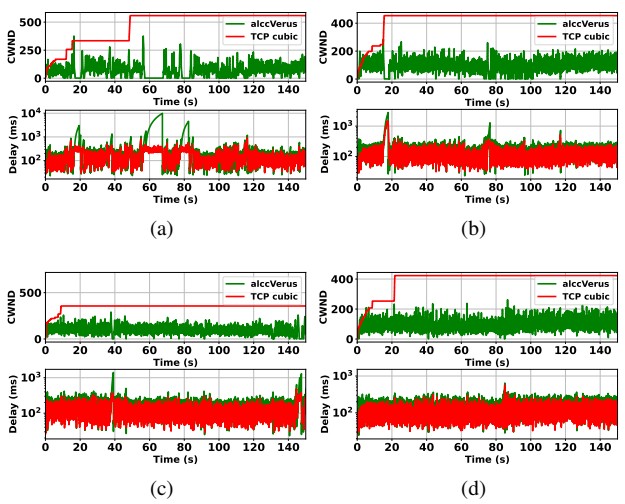

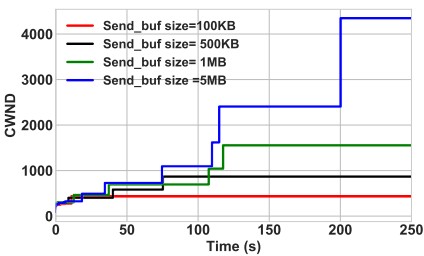

Figure 17: ALCC Verus and underlying TCP Cubic CWND analysis over a 4G network.

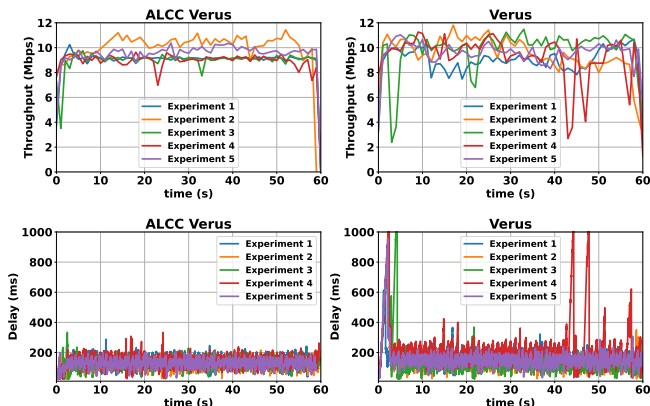

Figure 19: ALCC Verus atop Cubic Vs. Verus throughput and delay comparison in a real 3G cellular network.

Figure 18: TCP CUBIC's flat CWND behavior with different TCP send buffer sizes.

## 5.8 ALCC under Packet Losses

We have not observed severe packet losses by testing ALCC in a real 3G/4G cellular network as shown in Figure 16 and 17. This is because, in cellular networks, most transport layer losses are concealed from TCP by the underlying Radio Link Control (RLC) and MAC layers' re-transmissions and are further reduced by excessive buffering at the cellular base stations. However, we further investigate the ALCC's CWND behavior in a lossy channel by enforcing statistical packet losses via the MahiMahi network emulator. A 4G Verizon channel trace is used with an enforced 1% stochastic loss, a high error rate in cellular contexts, especially after the lower-layer recovery within the cellular network [12] [24]. Understandably, the underlying TCP congestion window is a bottleneck for ALCC, and ALCC can not send beyond the CWND offered by the underlying TCP. This is an explicit limitation of ALCC. Figure 20 shows that ALCC could not achieve similar performance to the native CC protocol in a lossy packet channel. This is because the underlying TCP cubic drastically reduces its CWND during a packet loss. Even if ALCC congestion control logic operates differently in

handling packet losses, it is not permitted to control congestion due to the restrained CWND allowed by the underlying TCP. As a result, ALCC follows the same behavior of the underlying TCP.

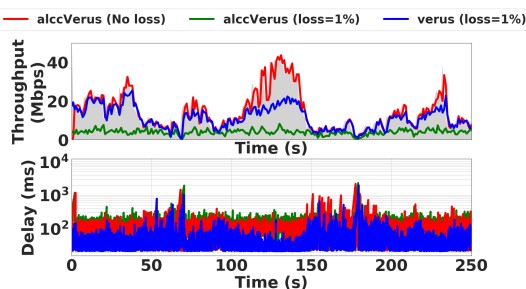

Figure 20: ALCC Verus Vs. Verus Throughput a and delay comparison with no loss and 1% enforced statistical packet loss

### 5.8.1 Potential Solutions

The idea of setting a high CWND for the underlying TCP via a dynamically Loadable Kernel Module (LKM) could be one potential solution to the above challenge. However, alteration in the main kernel itself requires a great effort because of the sheer complexity of many shared data structures among the TCP and network stack. A subset of the TCP parameters can be easily adjusted in Linux by using tools like *sysctl*. However, it is challenging to restrict the kernel's default TCP congestion control protocol to maintain a static high CWND in real-time. TCPTuner [33] is one such software package that may be utilized as an effortless bind. TCPTuner is a pluggable CUBIC congestion control module that allows congestion parameters control via a Graphical User Interface (GUI). The two control parameters α and β affect the CUBIC growth function, where α controls the increase in CWND and β

regulates the multiplicative decrease in CWND upon a loss event. Although β in the current implementation can not be set to 0 (which means an 0% reduction in the CWND). It may be assigned to a minimal value to maintain a CWND large enough for ALCC to operate smoothly, even in a lossy environment.

In general, the sysfs filesystem provides an interface to kernel data structures. The sys/module/ subdirectory contains one subdirectory for each module loaded into the kernel. Therefore, sysfs is the doorway to access all the writable parameters of any pluggable congestion control module. We seek to adopt similar methods described above to design an efficient loadable kernel module to assist ALCC by keeping the underlying TCP CWND to a high static value as part of our future work.

## 5.9 ALCC over VPN Tunnels

We investigate the performance of ALCC and native protocols with competing flows which may or may not use ALCC over a real-world VPN tunneling scenario. Figure 21 shows the experimentation setup. A Linksys dd-wrt router with an integrated VPN server is used, where two separate external BFPTD servers were connected to the router. The client connects to the VPN server over the Internet, creating a TCP tunnel.

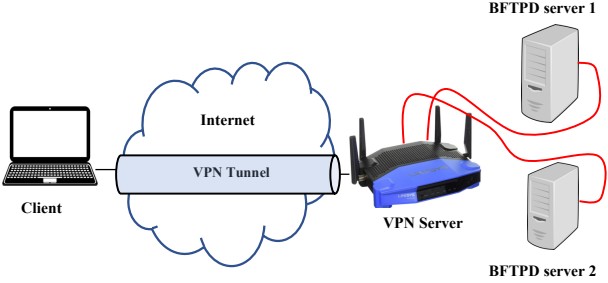

Figure 21: A VPN tunnel scenario between client and server(s)

**ALCC with competing TCP flows:** The experiment involves independently evaluating the performance of ALCC Copa (atop TCP Cubic) and native Copa (natively implemented over UDP) when sharing a tunnel bandwidth of 40Mbps with a competing TCP flow. These experiments investigate the protocols' fairness/friendliness to a competing TCP flow in a tunneling scenario. In the experiment, the TCP flow enters and exits the tunnel following an On/Off pattern, where it co-exists with an ALCC Copa or Copa flow for random periods. Figure 22 shows that ALCC Copa flows—depicted in red—are fair to competing TCP connections—depicted in blue—over the tunnel compared to the native Copa flows—displayed in green.

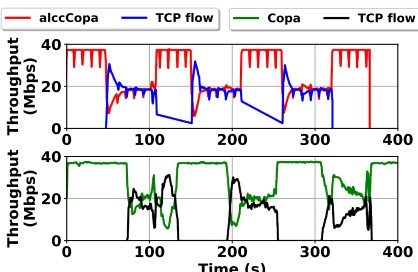

Figure 22: ALCC Copa and Copa Performance comparison over a tunnel with co-existing TCP flow.

**ALCC with competing UDP flows:** We further analyzed ALCC performance over the tunnel with co-existing UDP flows. The data rate of the UDP flow is restricted at each ingress in the tunnel. The aim is to examine the dominating nature of UDP flows when sharing bandwidth with competing ALCC flows. In the experiment, UDP flows are restricted to follow an On/Off pattern with varying data rates between 10Mbps and 30Mbps. Figure 23 and 24 show that the native UDP flows over the tunnel are not fair to competing TCP flows—not sharing the bottleneck bandwidth equally. It is observed that the UDP flows dominate the tunnel by sending at its maximum data rate at each ingress, taking a large fraction of the tunnel bandwidth.

The key finding from this experiment is that congestion control protocols built on top of UDP exhibit an unfriendly nature over the tunnel when competing for bandwidth with co-existing TCP flows from the same host. The ALCC implementation currently supports congestion control for a single flow from the end-host. For multiple ALCC flows from the same host, a congestion control manager is required for fair bandwidth sharing. We consider this as an essential extension of the ALCC framework in the future.

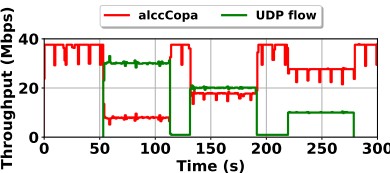

Figure 23: ALCC Copa Vs. UDP flow over a tunnel.

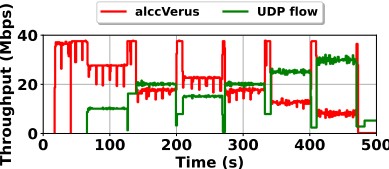

Figure 24: ALCC Verus Vs. UDP flow over a tunnel.

# 6 Related Work

Over the years, numerous TCP versions have been proposed [7, 8, 11, 16, 23, 27, 44, 46, 48] to optimize transport protocol performance. The default TCP Cubic revises the additive increase, multiplicative decrease (AIMD) practice of TCP to promptly saturate the link in high bandwidth-delay product (BDP) networks. But Cubic is inefficient to trace time-varying wireless link capacities properly. However, delay-based congestion control protocols, such as TCP Vegas [7], have earned increased attention in the context of cellular networks due to its performance concerning non-congestion induced losses.

Several other protocol designs focus on learning optimal performance, despite variations in the network environment. The essential approach is to search for actions directly to maximize throughputs and reducing delays. Remy [49] employs off-line training to achieve the optimal mapping connecting network conditions and the CWND adjusting function. In contrast, the performance-oriented congestion control (PCC) protocol [14] utilizes online learning to determine the sending rate for maximizing the value of a utility function based on feedback from the receiver in real-time. PCC Vivace [15], which followed after PCC [14], leverages ideas from online (convex) optimization in machine learning to do rate control while alleviating the bufferbloat problem. However, Vivace's performance in LTE networks suffer due to noisy environments. 3G and LTE network measurements [25, 38] have demonstrated that variations in the physical properties of the radio channel can cause significant performance differences. Due to highly variable channel fluctuations, cellular networks often use deep buffers, which leads to significant self-inflicted packet delays due to bufferbloat. Bufferbloat can be avoided by employing Active Queue Management (AQM) schemes like CoDel [37] and PIE [40], however, despite achieving low packet delays, these schemes suffer from under-utilization of link capacity. Without AQM, Cubic and NewReno rely only on packet drops as a sign of congestion. With deep packet buffers, this signal is too infrequent for active adaptation to varying link conditions. Protocols like Sprout [50] and Verus [51] overcome the spareness of packet drops by utilizing RTT and send/receive rate information with prediction strategies to conclude accessible link capacity. Sprout [50] is designed for real-time streaming applications that demand high throughput and consistently low packet delays over cellular networks. Verus [51] is a delay-based congestion control protocol designed for highly fluctuating mobile networks. Verus makes decisions on changing delay profile curve over time and adapts to the instantaneous properties of the channel conditions. BBR [10], also proposed by Google has shown good results over cellular networks. BBR uses the round trip propagation time and bandwidth of the bottleneck link to find the optimum operating point for CC. UDP based data Transfer Protocol (UDT) [22] is a user-space framework that carries the protocol design, used for configuring and evaluating new CC

algorithms. Unlike ALCC, UDT is a UDP-based approach that employs a CC algorithm targeting shared networks.

# 7 Conclusions

This paper was motivated by the difficulty we face in deploying new CC protocols for cellular networks. Despite significant advances in CC research for cellular environments, mobile applications are deeply wedded to the legacy TCP stack due to a variety of factors. This paper describes the design and implementation of the ALCC framework to enable migrating CC protocols to the application layer and derive similar performance to the native protocol on top of the legacy TCP stack. We have demonstrated the efficacy of ALCC by integrating three different recent CC protocols (Verus, Copa and Sprout) and showing that the ALCC version of these protocols have very similar performance characteristics (i.e. throughput and delay distributions) achieved by their original versions. We have also demonstrated how easy it is to integrate any CC protocol into ALCC, as well as to incorporate the ALCC framework into existing applications.

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
