# OpenReview forum: "ALCC: Migrating Congestion Control to the Application Layer in Cellular Networks"
_JSYS/2021/May_Papers — Submitted to JSYS May 21_

### Official Review · Reviewer_CnU1 · 2021-06-05
**very interesting paper with some unconvincing claims**

**Decision:**

Weak reject: interesting papers with flaws, not sure if they can be fixed in three months

**Review:**

Thank you for submitting your paper to JSys! Overall I really like the idea of implementing the congestion control algorithms within the application layer to eliminate the overhead of deploying new algorithms. The paper is also interesting in

Pros:
+ interesting idea and pragmatic solution
+ thorough evaluation with three well-known congestion control algorithms

Cons (explained later):
- implementation cost in kernel
- limitations in the scope of deployable algorithms

I'm little confused about the implementation of ALCC, similar to the reviewer 48B from Eurosys. From my understanding, ALCC still need kernel patches (as introduced in Section 4) to realize its function. I understand that such a patch is much more lightweight than implementing a new cc from the scratch. However, now that the kernel still needs modification (no matter a large patch or a small one), what's the difference for operators?

A suggested experiment: what if fix the congestion window to the maximum value and fully let the application layer takes over the control? I'm wondering if (and how much) the underlying cubic is still effective when the ALCC works above.

Besides, a key assumption in this paper is that the cubic always maintain a higher cwnd than other algorithms, which needs more convincing analysis. The authors try to explain that cubic *usually* maintains an unnecessarily high cwnd, which is different from *always*. Since the performance difference between algorithms usually depend on their actions in these corner cases, it would be better if the authors could provide more convincing arguments (theoretical analysis, or the detailed investigation on cwnds) on this.

My understanding of the assumption above is that cubic is the most *performance-oriented* algorithm compared to the three algorithms evaluated in the paper. However, there are indeed some other algorithms that are more aggressive in throughput than cubic (e.g., bbr or a finetuned pcc towards throughput). It would be better to discuss and clarify the scope of ALCC.

Other suggestions:
- Since ALCC controls the deliver of packets at the application layer, introducing such a new queue at the application layer might increase the end-to-end delay. It would be better if the authors could explain or measure such influence in detail.
- I fully agree with the authors that ALCC could ease the efforts for operators to implement new congestion control algorithm. However, instead of human work days, it would be more straightforward if the authors could present other quantitative metrics (e.g., line of codes).
- Another thing I don't understand is related to the zero-byte claim in the paper. How could the newly added header (introduced in Section 4.3) be zero-byte? If ALCC is encapsulating / decapsulating within the protocol stack, will such operators increase the overhead? If not, will such a header be delivered over the network?

**Expertise:**

Published in this area in the last 5 years

**Useful:**

yes

**Writing:**

Writing is beautiful, paper is easy to understand

---

### Official Review · Reviewer_ypqM · 2021-06-09
**Simple and reasonable idea. But there are many design and implementation problems.**

**Decision:**

Weak reject: interesting papers with flaws, not sure if they can be fixed in three months

**Review:**

Thanks for submitting the paper to JSYS. This paper presents an interesting system ALCC which implements congestion control for cellular networks in the application layer. I like the idea of moving congestion control to user space, which can significantly accelerate the adoption of new congestion control algorithms. However, I have the following concerns:

1. The authors implicitly assume that cwnd of kernel TCP (e.g., Cubic) should be larger than cwnd of application layer, which ALCC wants to enforce. Thus we have cwnd of ALCC = min(cwnd of kernel TCP, cwnd of application). This assumption may not always hold in practice, e.g., Cubic significantly reduces the congestion window due to small switch buffer and non-congestion packet losses (e.g., bit errors, failures).

2. It seems to me that ALCC lacks a general method to support customized ACKs required by various cellular network congestion control algorithms. The solutions in 4.1.3 seem very ad hoc to me. I worry that this can limit the scope of congestion control algorithms that can be supported by ALCC.

3. ALCC provides different APIs compared to legacy socket. This introduces non-trivial modifications to migrate an application to ALCC.

Here are my suggestions:

1.  Instead of assuming kernel TCP's window is always large enough, I suggest the authors enforcing a static high window size in kernel TCP, thus 'disabling' kernel TCP congestion control.  This should be easy to implement using a loadable Linux kernel module, like what you do in 4.1.2.

2. Instead of exposing customized APIs to applications, I suggest the authors using LD_PRELOAD to translate legacy socket APIs to ALCC APIs.

3. I hope the authors can summarize how to support customized ACKs in a more principle way. Currently 4.1.2. and 4.1.3 seem very unclear and ad hoc to me.

4. I like your observation that Android imposes restrictions on modifying kernel in section 4.3. But the solution described in 4.3 is not so related to 4.1 and 4.2, and lacks many important details.  I suggest the authors moving 4.3 to discussion.

5. Client/Server seem a little bit confusing as both client and server can send and receive data. Can you use Sender/Receiver instead?

6. There are many high performance user space network stacks designed for cloud environment. Many of them can support legacy socket APIs without requiring any application modifications, e.g., Mellanox LibVMA. Why not port them to cellular devices and implement congestion control algorithms on the top of them? I suggest the authors discussing these solutions in the paper.


**Expertise:**

Follow the literature for the broad topic of this paper

**Useful:**

yes

**Writing:**

Only experts can understand the paper

---

### Official Review · Reviewer_3y66 · 2021-06-09
**Review for ALCC**

**Decision:**

Weak reject: interesting papers with flaws, not sure if they can be fixed in three months

**Review:**

**Paper summary**

The paper proposes and implements ALCC, an application-layer framework that facilitates the development and deployment of congestion control (CC) algorithms for cellular networks on top of a legacy TCP stack. This is feasible due to the observation that TCP's congestion window is kept unnecessarily large on cellular networks, allowing ALCC the latitude to operate at a lower effective congestion window (or sending rate) than the underlying TCP. In the evaluation, this paper demonstrates that porting three well-known CC protocols -- Verus, Copa, and Sprout -- to ALCC requires little effort, while it is possible to maintain comparable performance as their native implementations at the same time.


**Strengths**

- This paper identifies an interesting research opportunity and comes up with a simple idea that works surprisingly well.
- The solution is practical and has the potential to accelerate the adoption of new CC schemes on cellular networks.
- Nice discussion of the motivations and clear comparison with QUIC and CCP.


**Weaknesses**

- Shallow and repetitive evaluation that has only scratched the surface of the problem and the solution.
- Missing evidence from the real world.
- The core idea sounds too good to be true and lacks justification, i.e., the paper does not describe in what scenarios ALCC is not applicable, and it is unclear when the underlying TCP congestion control will affect the efficacy of ALCC.


**Comments for author**

Thank you for submitting the paper to JSys! It was a pleasure to read it, particularly the sections before the evaluation.

One of the biggest strengths of the paper is to bring an elegant but unexploited idea to light and embed the idea into a practical framework, which may simplify the development of cellular CC protocols and potentially accelerate their adoption in real settings. I appreciate that you intentionally mimic the widely used socket APIs, as well as the effort of porting three CC protocols to your framework.

However, the paper is not yet in good shape unfortunately.

The obvious missing piece is an in-depth evaluation that reveals the internal mechanism. The paper shows that three CC protocols (Verus, Copa, and Sprout) achieve similar performance with and without ALCC, using repetitive experiments and figures. The entropy is low and the effective information conveyed to the audience is limited. By contrast, when things get interesting on lossy links where the performance discrepancy starts to manifest (Figure 13), the paper stops delving into it. Additionally, the micro-experiments for validating the design choices are largely missing (described as suggestions below).

Here are my suggestions:
- Plot how the underlying TCP CWND varies over time along with each cellular CC scheme's CWND (if available; plot sending rate otherwise). Show how often ALCC receives blocking signals from the underlying TCP and other noteworthy interactions between them.
- Report the performance numbers of TCP Cubic as a sanity check. Prior work has pointed out that even on cellular links, Cubic is not necessarily worse than these new protocols.
- It is better to include an experiment to illustrate that when there is no bufferbloat, ALCC's approach may not attain superior performance, which necessitates the setting of bufferbloat or cellular networks.
- Besides stochastic packet losses, what are the other scenarios (e.g., high-speed networks?) when ALCC-based CC differs from the native implementation? For these scenarios, give evidence to explain the unusual behavior, such as why ALCC-based Verus in Figure 13 achieves higher throughput than Verus between 100-150 seconds. Describing a hypothesis without a quantitative proof ("ALCC indirectly reduces the chances ...") or using words like "surprisingly" does not help shed light on the phenomenon.
- Even under the setting of stochastic packet losses, I doubt ALCC-based CC algorithms can always reach a higher throughput as shown in Figure 13. E.g., if an algorithm were informed a 10% random packet loss beforehand (on a link with infinite bandwidth), then it could leverage FEC and send 10% extra redundancy packets to maintain a high goodput. However, ALCC would have limited its sending rate to the underlying TCP's data rate, which would be extremely low under such a high loss rate. Therefore, the results of Figure 13 might not be generalizable.
- A cellular CC protocol implemented on a UDP socket typically also specifies a retransmission mechanism. Now that the retransmission has to be done by the underlying TCP, the implications are unclear to the audience. The paper would have offered more insights if it could present the implications and how ALCC could correctly avoid packet losses / retransmissions in certain circumstances.
- Protocols such as Sprout are intended to be used with low-latency applications such as videoconferencing, so it might be a good idea to replace the obsoleted RTMP streaming application with that. On a related note, the paper only describes how easy the integration with off-the-shelf applications is, while missing the impact on their performance after the integration.
- The three different ways to realize ALCC require a quantitative comparison. And the CPU usage is missing for the client/server implementation and the mobile Java library; even if it is infeasible to measure the CPU overhead on an Android phone for Copa or Verus, the paper could have reported the overhead compared with running the default kernel TCP.
- ALCC library is currently implemented as several threads -- what are the alternatives and pros/cons?

Apart from the above suggestions, real-world experiments are essential as well. Figure 1 and 2, the only figures that justify and motivate the core idea, are performed in a controlled environment. However, the evolution of CWND can be much more complex on real cellular networks. Empirical results have shown that TCP exhibits drastically different behavior on different operators' networks, e.g., the bufferbloat problem is less prominent sometimes, which might invalidate ALCC or negate the performance gains of ALCC. There should be real-world experiments too in the evaluation section; they need to be performed repeatedly with error bars included in the results. Real experiments would make the paper's arguments and findings more compelling.

*Other comments:*

- Why are we not seeing the sawtooth pattern of TCP Cubic in Figure 1? Is it because a large bin is used in the graph?
- It does not make sense to require developers to replace their calls to send() with alcc_send() in every place, so I wish a more elegant solution is implemented, such as the LD_PRELOAD trick mentioned in Section 4 or system call hijacking (which understandably has other limitations).
- Thank you for the replies to my comments! I agree that there is value in the server-side ALCC library, but it still sounds possible and straightforward to implement Sprout on top of a non-blocking TCP socket without the need of ALCC's client/server library, so long as it operates at a lower speed than TCP with the help of TCP's blocking signal. The current paper draft does not compare with the alternative approach, or describe what else ALCC's client/server library actually does beyond a simple user-space wrapper around a TCP socket. Besides, the authors did a better job in their replies explaining the interactions between ALCC and the underlying TCP than the paper does; more experiments to break down the impact are required to support these interaction claims regardless.

**Expertise:**

Actively publishing in this area

**Useful:**

yes

**Writing:**

Non-experts can understand the paper

---

> ### Comment · Reviewer_3y66 · 2021-11-12
> **Review for the revision**
>
> Thank you for the great effort of adding real-world experiments! It significantly adds value to the paper and demonstrates ALCC's applicability.
>
> The revision has addressed most of my concerns, and I only have a few minor ones:
> - Section 2.1.2: "QUIC has not yet reached a stable RFC" -- QUIC is now RFC 9000.
> - Figure 1: Your response says that we are not seeing the sawtooth pattern because there are no losses in the network, which I agree but does not answer my question. The cwnd should keep increasing given no losses but it stops at around 100 packets, which seems too soon to hit the receiver window or anything. This figure is not a deal breaker, but please double check and reply with a correct explanation.
> - Figure 16: Thanks for including this figure but I am a bit confused about the lower part of it. Since ALCC Verus runs atop TCP Cubic, each application packet will wait in the ALCC buffer first until the underlying TCP socket accepts it. Therefore, shouldn't the packet delay of ALCC Verus be always higher than (or equal to) TCP Cubic?

---

> > ### Author Response · Authors · 2021-11-18
> > **Response to the revision review comments**
> >
> > Thank you for the feedback and valuable comments. Here we reply to the minor comments you raised above:
> >
> > - Thank you for highlighting that QUIC has a stable RFC now. We will update section 2.1.2 accordingly.
> >
> > - Regarding the sawtooth pattern of TCP and why it is not increasing beyond a certain limit. TCP CWND experiences a flat-static CWND due to having no packet loss observed by the TCP sender. The main reason behind this is the large buffers used in cellular networks and the lower link-layer retransmission mechanism that hides losses from TCP. In our experiments, we used the default MahiMahi settings (which use an infinite buffer) to approximate the behavior of a real cellular network. An infinite buffer would yield the same effect as in the cellular networks where no packet losses will be observed. The static CWND seen in Figure 1,4, and 5, is referred to as the "Flat TCP". This is well documented and studied by Jiang et al., in their paper "Understanding bufferbloat in cellular networks" published at SigComm 2012 (https://dl.acm.org/doi/abs/10.1145/2342468.2342470).  They have dedicated a subsection, "5.1 An Untold Story", to explain this. In addition, we did a simple experiment with a finite buffer of MahiMahi, and the TCP CWND had the expected sawtooth behavior.
> >
> > - Regarding Figure 16, you are absolutely correct. ALCC delay should always be higher or equal to TCP's one. However, these two delays are not recorded/measured the same way and have different recording resolutions (sampling). TCP's delay is measured from the experiment pcap file, where we used Wireshark's "tcp.analysis.ack_rtt" to estimate these delays. The delay, i.e., RTT, is calculated by Wireshark on packets that have ACKs of past segments and is calculated as the time delta between the original packet's SEQ and this packet's ACK. For ALCC delays, we calculate those at the ALCC application layer by maintaining the timestamps of every ALCC packet sent and then taking the time difference when the ACK of that packet is received. As such, ALCC's delay resolution is much finer than TCP's because in ALCC we measure the delay for every sent packet. On the other hand, the TCP delays calculated by Wireshark have lower sampling given that TCP does not ACK every packet, and it tends to send cumulative ACKs back. That's why the figure sometimes shows lower values for ALCC, given that it's recorded at a finer resolution than TCP.

---

> > > ### Comment · Reviewer_3y66 · 2021-11-18
> > > **LGTM**
> > >
> > > "5.1 An Untold Story" indeed attributes the static CWND behavior to receive window, although the max tcp_rmem value in your experiment is still surprisingly small (Linux's current default is 6 MB).
> > >
> > > My question about Figure 16 is properly addressed. I have no more comments. Thanks for your reply!

---

### Official Review · Reviewer_8yrc · 2021-06-12
**This paper presents an approach to implementing congestion control at the application layer. The key novelties are the approach of using large windows at TCP layer to support free window changes controlled purely from the app layer, without datapath modifications. The paper shows that the app-layer implementation has high fidelity with respect to implementations over UDP.**

**Decision:**

Weak accept: good paper with flaws that can be fixed in three months

**Review:**

Thanks for submitting this paper! I enjoyed reading it.

summary of the paper
====================

This paper presents an approach to implementing congestion control at
the application layer. The key novelties are the approach of using
large windows at TCP layer to support free window changes controlled
purely from the app layer, without datapath modifications. The paper
shows that the app-layer implementation has high fidelity with respect
to implementations over UDP.

strengths
=========

(1) I think it is a really interesting idea to piggyback protocol
    windows at app layer over large windows at TCP layer. In scenarios
    where the congestion windows can be kept large indefinitely, I
    think it makes a lot of sense.

(2) Real implementation on linux and android. I really enjoyed reading
    the descriptions in 3.2 and section 4 on how the userspace
    components can read tcp_info for congestion window information and
    how netlink/iptables is used to intercept ACKs.

(3) Not requiring datapath modification is a very useful feature in
    ossified environments like cellular backbones and mobile OS
    ecosystems.

weaknesses
==========

(1) The applicability of the framework seems limited.

- Purely-server-side modifications are already possible today with
  large deployments -- think schemes like BBR and algorithmic
  innovations like TLP, RACK etc. that were unilaterally deployed on
  server side. So, downlink transfers (ex: web server -> mobile UE)
  are already doable if no explicit receiver feedback is required.

- the connection must be long-lived, allowing the TCP cubic window to
  rise sufficiently high. So this approach won't work for
  latency-sensitive apps that only send a few packets.

- if a cellular link exhibits highly variable RTTs or displays
  otherwise lossy behavior, the TCP cubic window itself might be low,
  which upper bounds what the ALCC algorithm can do in terms of window
  adjustments. This makes it difficult to build ALCC protocols that
  respond to such scenarios more effectively than tcp cubic.

- Reliance on UDP/QUIC looks poised to accelerate in the future, as
  would likely developments in the flexibility allowed by datapaths
  (e.g. Android TCP). ALCC is certainly a good point solution but its
  shelf life appears limited.

(2) The paper must report full CPU overhead measurements rather than
    just relative overheads. Going from 5% to 7.5% looks very
    different from 50% to 75% even though both are 1.5x increase. Also
    claiming "comparable cpu overheads to ccp" is dubious given that
    the ccp overheads were obtained at 10 Gbit/s

    If possible, please also provide measurements from a mobile phone
    client as well by implementing a UDP-socket based version of the
    protocols.

(3) Showing more protocols would help make the point about
    expressiveness of the framework (BBR, Vegas). Specifically, the
    eval seems to lack purely pacing-based protocols like BBR.

(4) The comparisons to ccp and quic need to be refined.

    CCP does not require total rewrites of CC algorithms. Further,
    some datapaths already have CCP support mainlined, like the mvfst
    QUIC library from Facebook and the mTCP DPDK-based TCP
    implementation.  Support for the kernel datapath (without
    upstreaming) is now easier than ever with bpf-based alternatives
    https://lwn.net/Articles/809092/

    Also, migration to QUIC on mobile client is doable without
    invasive app changes. For example, uber recently managed to
    migrate their app to quic, their experience suggests not requiring
    extensive app modifications.
    https://eng.uber.com/employing-quic-protocol/

(5) The authors must describe their mobile app design and interaction
    with congestion control more thoroughly.

    Specifically, it is unclear how the http library enforces
    congestion windows, and measures RTT estimates, etc. required for
    CC algorithms. An HTTP req/resp is not the same as a single TCP
    packet/ACK, since http responses may straddle multiple packets,
    for instance.

    Is MDI a prerequisite to run ALCC for uplink in mobile clients?

    To me, not requiring the mobile client to change its OS for uplink
    data transfer (mobile -> server) is one of the key novelties of
    ALCC's design. But the description of the solution is incomplete.

Detailed comments
=================

intro / 1:

- The descriptions make it appear as if TCP is inescapable, but there
  are of course apps out there which use HTTP over UDP, QUIC, etc.

- how can ALCC use a blocking signal from the network stack since
  processes are put to sleep upon blocking (with blocking socket)? Or
  is the point that the socket is blocking from the point of view of
  the app, but always non-blocking from the point of view of ALCC?

- I didn't understand the second point about the blocking signal "The
  blocking signal also affects the behavior of the higher layer
  protocols to adapt to the varying signal."; maybe give a concrete
  example.

- The comment that "ccp requires users to re-think congestion control
  algorithms entirely" seems unfair. It is possible to use a simple
  "on-datapath" component and implement most or all of the algorithm
  in the userspace component.

- it is not very surprising that almost any TCP modification in the
  downlink diretion can be supported with just server-side
  modification, unless specific client-side feedback is required.

- what do you mean by zero-byte overhead? modifications to tcp
  congestion control don't (typically) require packet structure
  modification.

2

- "The last hop from the VPN server to the desired remote host/server
  will continue to rely on the host’s TCP variant, thereby negat- ing
  performance gains." won't this be true for ALCC protocols as well?

- what does it mean for "other network entities to participate in the
  CC"?

2.1:

- "UDP lacks required security support" is simply untrue given
  alternatives like QUIC and DTLS

- QUIC deployment numbers are trending up, how significant still is
  middleboxes dropping UDP?

  https://datatracker.ietf.org/meeting/106/materials/slides-106-maprg-quic-deployment-update-00

- 2.1.1:

- modifying the "datapath", not data

- in general, comparisons against CCP felt more like engineering
  concerns rather than conceptual design differences. The main
  conceptual difference appears to the removal of the requirement to
  modify the datapath, which the authors should boldly say outright!

- in general, it is not clear how ALCC will receive signals from the
  TCP stack without modifying the TCP stack itself or at least have
  some measurement hooks. Is what is available in tcp_info enough
  already?

- architecturally, datapath modifications are mainly a problem on
  mobile clients. They aren't an issue on the server side since
  running new variants or tuning algorithms on server side is fairly
  common practice.

2.1.2: alcc vs. quic

- "why there are no successful attempts to include any of the newer
  protocols in literature over QUIC" -- support for ccp in mvfst
  library was upstreamed after the papers referenced here were all
  published. It is fairly easy to implement CC algorithms.
  https://github.com/facebookincubator/mvfst/tree/master/quic/congestion_control/third_party/ccp/ccp_generic_cong_avoid/src

- how significant is the issue of quic packets being dropped by
  middleboxes at this point? youtube and facebook are driving lots of
  quic traffic, and if the trends persist, operators will be pressured
  into supporting quic.
  https://datatracker.ietf.org/meeting/106/materials/slides-106-maprg-quic-deployment-update-00

3.1:

- it would be interesting to consider a scenario where the link is
  lossy with a highly variable RTT.

3.2

- using tcp_info in user space to get feedback from kernel is pretty
  neat.

- "if the TCP socket reports a full buffer and blocks on a potential
  transfer (or indicates full buffer in a non-blocking socket), ALCC
  delays the next packet transmission. "  -> how would a blocking
  socket "report" anything, since the process and the ALCC library
  would actually be put to sleep?

- an architecture figure or a forward-pointer to one will help in this
  section

4.1.1

- what purpose do the ACKs from the kernel module serve?

- does your framework support other operations on the socket, like
  sockopt? In general, replacing operations with other calls is fine
  so long as the semantics of the socket object remain similar and the
  rest of the app can just assume the prior semantics (before ALCC
  interposition).

- what's the return value of the send() call? is it the amount of data
  pushed into the TCP socket (blocking semantics) or the amount of
  data in ALCC's circular buffer?

4.1.2

- netfilter hook to intercept ACKs and send to userspace module is a
  nice idea, to send protocol ACKs to the mobile client (for uplink
  data transmissions). But netfilter and iptables may not scale to
  large line rates or number of connections, especially on the server
  side. Scalability of these mechanisms will definitely be an issue. A
  different mechanism, like a shared memory buffer or running a BPF
  hook with maps, might turn out to be more CPU efficient. Have you
  considered evaluating your system against increasing number of
  concurrent connections?

- what if the userspace process fails? Having external hooks to
  register and deregister iptables rules seems like a problematic
  design from the fault tolerance perspective.

4.1.3

- The discussion of the verus implementation seems to indicate that
  substantial datapath modifications were in fact required even for
  ALCC server-side implementation. do you anticipate that the set of
  datapath features is frozen at this point, or that you will need
  more based on trying out new algorithms?

- Copa description seems a little in the weeds.

4.2

- if the client needs to send an ACK for feedback, that seems to go
  against "zero byte overhead" mentioned in the intro?

4.3

- implementation seems to use OkHttp, which won't work for any app
  that doesn't use okhttp or uses more generic socket API-based
  communication. This is a weakness of the implementation.

- adding a custom HTTP header is certainly a nice way to not require
  kernel modifications for things like RTT measurements. But HTTP
  messages can span multiple packets, especially responses. How is
  this handled?

- didn't follow the discussion on why MDI is needed. It seems like an
  orthogonal approach to implementing the CC.

- Notions of window sizes, ACK feedback... seem pretty hard to
  control/obtain from the http layer? you might never receive an
  app-level response from a partial http request and you're not
  getting any ACKs or socket interface feedback from TCP either. how
  well does the control loop actually work?

4.4

- did you try integrating with existing mobile client apps?

5.1

- is measurement under low contention/low data rates sufficient? 2.5/5
  Mbps flows from the cell phones may or may not capture anything
  meaningful about the channel capacities. it would be nice if the
  paper clarifies what these traces are really useful for

5.2 / 5.3

- it would be useful to dig into why the average delays are higher
  with ALCC-verus, both in the median and the quartiles. Is it
  fundamental? same thing with sprout 5.3.

- there are no space constraints in this journal paper! :) feel free
  to report all the results from all the traces.

5.6

- please show the actual cpu overheads. 1.5x-2x is a significant
  premium? The numbers from ccp were reported at 10 gbit/s, showing an
  increase of a few percent cpu (e.g., 2% -- 5%), which is very
  different from running on a cellular trace which runs at tens of
  10mbit/s or lower.

- "Given that native Copa or Verus can not operate on the native
  Android mobile in the uplink direction, phone measuring the CPU
  overhead was not feasible."  -> what about sprout? also, we could
  still compare the overhead over just running tcp cubic, to get a
  sense of how much it increases.

- why not develop a version of the algorithms over udp, like they work
  on non-mobile clients?

- "1% a relatively high error rate in cellular contexts especially
  after lower-layer recovery within the cellular net- work [13]. " ->
  the uber blog post I linked above disagrees.


**Expertise:**

Published in this area in the last 5 years

**Useful:**

yes

**Writing:**

Non-experts can understand the paper

---

### Author Response · Authors · 2021-10-16
**Revised paper submission**

Dear Editor and Reviewers

Please allow us to submit the revised paper on Sunday. Unfortunately, the submission deadline given happens to be on a weekend. Just to clarify that Friday and Saturday is a weekend in the UAE. We have made all the relevant changes to the paper and it would be in its final shape by today. We request you to give us until tomorrow to submit the final version along with detailed responses to Reviewer’s comments.

Thank You

Kind regards

---

### Author Response · Authors · 2021-10-17
**Submission of revised paper and rebuttal document**

Dear Chairs and reviewers,
We have submitted our revised paper along side with a detailed rebuttal highlighting all the changes made by us to address the reviewer's comments.
Given that there was only one pdf document file submission allowed, we have combined the rebuttal with the revised paper in a single document starting with the rebuttal.

We look forward for your reviews and feedback.

---

> ### Comment · Area_Chair_hHWk · 2021-10-17
> **Confirmation**
>
> Dear authors, thanks for submitting the revisions. We will review it and get back to you in a month.

---

### Comment · Reviewer_CnU1 · 2021-11-12
**Satisfactory revisions with minor issues.**

Thank you for submitting the revised manuscript to JSys! Most of my concerns have been addressed in a proper way, and the additional experiments meet my expectation. Thanks for your valuable revision efforts! However, I do have another concern about the cwnd of cubic:

The authors have analyzed the variation of cwnd of the underlying CUBIC, thanks for your efforts! However, questions still come to my mind after checking the results of cwnd (Figure 1, 4, 5 in the revision letter). Why does the cwnd keep exactly the same in most of the time?
Even if the authors respond that the missing of sawtooth is due to no packet loss in the network, from the reviewer's understanding, cubic will keep increasing its cwnd every RTT, until the bottleneck buffer is full (causing packet losses). It's quite interesting to see that cubic holds a static cwnd. Is it rwnd-bounded?

I'd highly appreciate it if this could be explained better in the camera-ready version (if finally accepted).

---

> ### Comment · Area_Chair_hHWk · 2021-11-12
> **awaiting for authors' response**
>
> Authors are welcomed to respond to the reviewer's concerns, the earlier the better, as they may be shared by other reviewers.

---

> ### Author Response · Authors · 2021-11-18
> **Flat TCP behavior**
>
> Thank you for your comments and for bringing up this important point.
>
> TCP CWND experience a flat static CWND due to having no packet loss observed by the TCP sender. The main reason behind this is the large buffers used in cellular networks, as well as the lower link layer retransmission mechanism that hides losses from TCP. In our experiments we used the default MahiMahi settings (which use an infinite buffer) in order to approximate the behavior of a real cellular network. Having an infinite buffer would yield the same effect as in the cellular networks where no packet losses will be observed.
>
> The static CWND seen in Figure 1,4, and 5, is referred to as the "Flat TCP". This is well documented and studied by Jiang et. al., in their paper "Understanding bufferbloat in cellular networks" published at a SigComm workshop in 2012 (https://dl.acm.org/doi/abs/10.1145/2342468.2342470). Where they have dedicated a subsection, "5.1 An Untold Story", to explain this.
>
> In addition, we did a simple experiment with a finite buffer of MahiMahi and the TCP CWND had the expected sawtooth behavior.

---

> > ### Comment · Reviewer_CnU1 · 2021-11-18
> > **Good explanation**
> >
> > Thanks for the detailed explanation and the reference! From the Section 5.1 of the sigcomm workshop paper I can see that the cwnd is rwnd-bounded. Thank you!
> >
> > I'm happy with the answer and have no further concern.

---

### Comment · Reviewer_ypqM · 2021-11-13
**Good revisions with minor issues**

Thanks for submitting the revised version. I really appreciate the authors' efforts to address reviewers' comments. Most of my concerns have been properly addressed by the authors except for the following two:

1. I appreciate that authors use many experiments to show that TCP Cubic ramps up to a very high CWND, which ALCC exploits to perform congestion control—staying under the TCP radar. But I am not convinced that altering the CWND of TCP to a static high value with a loadable kernel module is not easy. It seems to me that you just need to implement several simple call back functions (e.g., .ssthresh and ..cong_avoid) and set initial window. In my opinions, this implementation is even simpler than your Netfilter based kernel module which intercepts every ACKs and copies them to user space CC. Therefore, I suggest authors at least discussing this method in the paper.

2. I understand that retransmission is handled by underlying TCP stack and section 4.1.3 mainly focuses on customized ACK mechanisms. Can you summarize how to support customized ACK mechanisms of different CCs in more details? Current 4.1.3 seems a little bit ad hoc and unclear to me.

---

> ### Comment · Area_Chair_hHWk · 2021-11-20
> **Author response needed**
>
> Dear authors,
>
> We are in the last round of iteration. Could you please respond to the above comment as soon as you can. Thanks!

---

> > ### Author Response · Authors · 2021-11-20
> > **response**
> >
> > Apologies for the delayed reply on the above comment. We just responded to the comment now. Thank you for the reminder.

---

> ### Author Response · Authors · 2021-11-20
> **Response**
>
> Thank you for your comments and valuable feedback.
>
> Regarding the 1st raised point, we would like to clarify that ALCC does no alter the CWND of TCP. In fact, ALCC does not set or reconfigure anything of TCP. The kernel module is used simply to send copies of the TCP acknowledgements to the ALCC user space, nothing else. The high static value of TCP comes naturally in cellular networks because of the large used buffers as well as the underlying link layer retransmission mechanisms that hides losses from TCP, thus forcing TCP to ramp the CWND. This phenomena will happen with and without ALCC in cellular networks.
> In addition, thank you for your suggestion on implementing simple call back functions, however we don't believe that this can be simply implemented. This would probably require changes in the Linux kernel to modify the default TCP behavior. Instead, our second implementation that relies on custom ALCC acknowledgements does alleviate the need for a Kernel module, thus simplifying the design and implementation.
>
> Regarding the 2nd raised point, the customized ACK mechanisms mentioned in section 4.1.3 is simply an application layer acknowledgment implementation. Where each ALCC packet will have its own sequence number, and ALCC would record the exact time when that packet (i.e., sequence number) was sent into the network. Then at the client side, for each received ALCC packet the client would send back an ACK with the corresponding sequence number. Upon receiving such an ACK at the ALCC sender, ALCC can calculate the round-trip-time by taking the time difference between the receive time and the stored send time. This RTT is then internally used by the CC algorithms, whether Verus or Copa, to update the sending rate. We thank the reviewer for highlighting that the description of this was not super clean and we plan to add more description on this in section 4.1.3.

---

> > ### Comment · Reviewer_ypqM · 2021-11-22
> > **Large static TCP window needs in-depth discussion**
> >
> > I agree that in most of cases that Cubic's window is very large, thus alleviating the need to enforce a large static TCP window. However, since TCP static window is an obvious and clean baseline, it requires an in-depth discussion on why exactly this is not easy to implement, for example, which parts of kernel code must be modified to enforce a static window.

---

> > > ### Author Response · Authors · 2021-11-23
> > > **Large static TCP window Kernel implementation**
> > >
> > > Thank you for the reply and the clarification on the comment raised. Regarding the complexity and feasibility of changing the Kernel TCP implementation to support a static large CWND size, the problem of such an approach is the following:
> > > - modifying the Kernel implementation of an OS is a very complex endeavor which require a lot of work and expert knowledge in navigating, understanding, and making the correct changes in the correct places.
> > > - Also such changes would also require a complete kernel recompilation which usually takes several hours.
> > > - These changes are not OS independent, meaning that it has to be done for every different OS and Kernel. This simply does not scale.
> > > - The idea behind ALCC was to provide a library that content providers can easily install as an additional application/module without having to recompile existing OS Kernels.
> > > - Changing the Linux Kernel would also violate the Linux update procedure, given that these changes has to be re-enforced every time after each Kernel update.
> > >
> > > If the paper is accepted, we will add some of these explanations into the paper.

---

> > > > ### Comment · Reviewer_ypqM · 2021-11-23
> > > > **Whether large static TCP window is doable using a loadable kernel module**
> > > >
> > > > In Linux, most of TCP congestion control algorithms are implemented as loadable kernel modules, such as tcp_cubic.c and tcp_dctcp.c. A loadable kernel module can be installed and removed during system running time without recompiling kernel. Hence, I do not treat it as kernel modification. ALCC also has a loadable kernel module to intercept ACK packets.
> > > >
> > > > My question is whether a large static TCP window can be realized in a loadable kernel module. If not, which parts of kernel code will be required to modify? I hope the authors can discuss such details in the paper.

---

> > > > > ### Author Response · Authors · 2021-11-23
> > > > > **loadable TCP kernel modules**
> > > > >
> > > > > Thank you for highlighting the above, you are correct that it seems to be possible to do changes to the tcp_cubic.c and simply recompile that modified module and then reload it into the Kernel. Unfortunately, we have not tried this before, so we can't simply assess how easy or difficult this can be, in addition to what side-effects that might cause.
> > > > >
> > > > > We fully agree with your last statement that this should be discussed in the paper. However, even if this is successfully implemented, we believe that it won't change the behavior of ALCC and its performance apart from ensuring that the lower TCP CWND is statically high. ALCC's would still require either its own Kernel module to share the TCP ACKs to the ALCC user space implementation, or ALCC having its own custom ACKs.
> > > > >
> > > > > We would also add this as part of our future work/outlook as an additional and easier way to assist ALCC ensuring that TCP will have a large static CWND.

---

> > ### Comment · Area_Chair_hHWk · 2021-11-22
> > **Clarification please**
> >
> > Dear authors,
> >
> > Just wanted to nudge you to give this comment more thought. Regarding #1, if I understand the reviewer's suggestion correctly, the point is that the idea of setting a static, high cwnd may appear to any reader (including most reviewers and myself) as an obvious and clean baseline. That's also why it was listed the very first item of the major revision requests. Your point is well taken that doing so involves some changes to the kernel, but it is not obvious why this approach is very difficult to implement (I share the reviewer's confusion here). To be clear, the question isn't whether this is a better solution than yours; instead, the point is that the readers need to understand exactly what would be involved in implementing this baseline as well as what the particular difficulties one would face if they try to implement the baseline.
> >
> > I appreciate the plan to incorporate #2 in the final draft. It fills a crucial missing piece in the technical presentation.

---

### Comment · Area_Chair_hHWk · 2021-11-24
**Accepted (with shepherding)**

Dear authors,

Sorry for the delay. I'm pleased to inform you that the manuscript is conditionally accepted, and the final paper will need to be approved by the shepherd. Please work with the shepherd to (1) include necessary details missed in the current draft, and (2) smooth the writing so that the new and old content forms a coherent flow.

---

> ### Author Response · Authors · 2021-11-26
> **Thank you**
>
> This is great news. We thank you and the reviewers, and we look forward to be working with the shepherd to finalize the manuscript.

---

### Comment · Reviewer_ypqM · 2021-12-07
**Camera ready version**

Dear authors,

I am very happy to work with you to finalize the manuscript. Can you please provide a plan to list all the changes you want to make in the final version? The changes should address review comments in both rounds.

Thanks for your patience.

---

> ### Author Response · Authors · 2021-12-08
> **Modification Plan for Camera Ready Version**
>
> Dear Chairs and reviewers;
>
> Thank You for your time and efforts in improving this manuscript. We want to update the manuscript by addressing all the concerns of the reviewers that are as follows:
>
> 1. Why does TCP CWND experience a flat, static CWND behavior even with no packet losses?
> 2. The idea of setting a static and high CWND for the underlying TCP via a loadable kernel module. And the pros and cons involved in such an implementation.
>
> 3. Summarizing section 4.1.3 regarding the support of customized Ack mechanisms for different CCs.

---

> > ### Comment · Reviewer_ypqM · 2021-12-11
> > **Good Modification Plan**
> >
> > The modification plan looks good to me.

---

> > > ### Author Response · Authors · 2021-12-23
> > > **updated pdf according to agreed modification plan**
> > >
> > > We have updated the pdf according to the agreed plan. How shall we submit that version?

---

> > > > ### Comment · Area_Chair_hHWk · 2021-12-24
> > > > **Please highlight the changes?**
> > > >
> > > > Dear authors, Thanks for the update.
> > > >
> > > > Could you highlight the new changes in a different color than the previous revisions?

---

### Comment · Reviewer_ypqM · 2022-01-19
**Good camera ready version with some minor issues**

Thanks for submitting the camera ready version. In general, this version addresses all the concerns raised by reviewers. I only have a few minor concerns.

1. The following statement in the introduction seems incorrect to me as ALCC provides customized APIs rather than standard socket APIs to the application layer. In addition, a period is missing between two sentences.

*ALCC exposes the same TCP Berkeley socket APIs to the application layer, making it very easy to integrate into existing applications For recently developed CC protocols [4, 50, 51], we demonstrate how easily these protocols can be blended into the ALCC framework and maintain the application layer congestion window in contrast to the underlying TCP congestion window.*

2. In section 5.7, it seems to me that it is underlying *TCP send window rather than congestion window* stays static after a specific time. In addition, it seems that the static window phenomenon is first shown in Figure 1. How about moving the explanation to section 3.1?

3. Thank you for adding 5.8.1 to discuss the alternative solution. But this section seems a little bit long to me with too many unnecessary details. Can you shrink it a little bit?

---

> ### Author Response · Authors · 2022-01-25
> **Minor issues corrected**
>
> Dear ypqM, thank you for you suggesting the above minor issues. To address them we have:
>
> 1.	The following statement in the introduction seems incorrect to me as ALCC provides customized APIs rather than standard socket APIs to the application layer. In addition, a period is missing between two sentences.
>
> We have now corrected the statement to:
>  “ALCC provides customized APIs which indirectly expose the same TCP Berkeley socket APIs to the application layer….....”.
>  Also we have added a period between the two sentences. Thank You for highlighting this.
>
> 2.	In section 5.7, it seems to me that it is underlying TCP send window rather than congestion window stays static after a specific time. In addition, it seems that the static window phenomenon is first shown in Figure 1. How about moving the explanation to section 3.1?
>
> We agree with your observation. However, the figures demonstrate TCP CWND’s flat behavior rather than the sending window. This is an interesting observation, and we consider investigating this phenomenon in our future work.
> We believe the explanation about TCP’s flat static CWND in Section 3.1 is too early and would divert the readers’ attention from the main ALCC logic. However, we have added a statement in section 3.1, which says, “An interesting observation in Figures 1 and 2 is the static behavior of TCP cubic CWND. This phenomenon is explained in detail in Section 5.7.”
>
> 3.	Thank you for adding 5.8.1 to discuss the alternative solution. But this section seems a little bit long to me with too many unnecessary details. Can you shrink it a little bit?
>
> We have now shrunk the explanation as suggested.

---

> > ### Comment · Reviewer_ypqM · 2022-01-26
> > **Investigate TCP static congestion window**
> >
> > Dear authors,
> >
> > Thanks for your reply. It seems to me that the static TCP congestion window phenomenon deserves some investigations. I think it is probably due to TCP send buffer limit. Can you try different TCP send buffer sizes to validate this hypothesis? If the hypothesis holds, we can add explanations in the paper. Otherwise, let's leave this as future work.

---

> > > ### Author Response · Authors · 2022-02-02
> > > **TCP send buffer size**
> > >
> > > Dear ypqM,
> > > Thank you for the reply and the suggestion. We have done a simple investigation by conducting multiple experiments with a bftpd server and a separate Linux FTP client that is setup to request a large file from the server over TCP Cubic. The CAPTCP Analyzer tool was used to log the CWND. We slowly increased TCP's snd_buffer with each experiment, starting with 100KB and then increasing it to 500KB, 1 MB, and 5MB buffer sizes. We can confirm that there is a direct correlation between the TCP snd_buffer size and the CWND reached during the flat phase. Please see the following plot of the TCP CWND sizes from the following link:
> > > https://www.dropbox.com/s/rmn783315zwmykz/snd_buffer_effect.pdf?dl=0
> > >
> > > We can add a sentence in the paper stating that we did some experiments confirming the correlation between the snd_buffer and the TCP CWND flat value.
> > >
> > > We hope that we have now addressed all the reviewers' comments and suggestions.

---

> > > > ### Comment · Reviewer_ypqM · 2022-02-06
> > > > **Re: TCP send buffer size**
> > > >
> > > > Thanks for your efforts to confirm this. In current version, it seems to me that the explanation of static congestion window is in second paragraph of section 5.7. Can you update the explanation and highlight changes?

---

> > > > > ### Comment · Area_Chair_hHWk · 2022-02-08
> > > > > **Re: TCP send buffer size**
> > > > >
> > > > > Dear authors,
> > > > >
> > > > > Just wanted to clarify, please add in the paper the new text describing the latest experiment and the conclusion.
> > > > >
> > > > > Thanks!

---

> > > > > > ### Author Response · Authors · 2022-02-08
> > > > > > **TCP CWND and send buffer size correlation**
> > > > > >
> > > > > > As advised, we have now added the additional experiments conducted to confirm TCP CWND correlation with TCP send buffer size. Please find the edited text highlighted in blue. Specifically in Section 5.7.
> > > > > >
> > > > > > Thanks

---

> > > > > > > ### Comment · Reviewer_ypqM · 2022-02-08
> > > > > > > **Confusing explanations in section 5.7**
> > > > > > >
> > > > > > > Dear authors,
> > > > > > >
> > > > > > > It seems to me that section 5.7 gives two explanations to the static congestion window phenomenon: receiver buffer of the peer side and send buffer. The first explanation does not seem reasonable to me.  While the receiver buffer can limit the send window, but it should not have any impact on congestion window.

---

> > > > > > > > ### Author Response · Authors · 2022-02-11
> > > > > > > > **Fixed the writing of 5.7**
> > > > > > > >
> > > > > > > > Dear ypqM,
> > > > > > > > We have edited the text at the end of 5.7 to reiterate on the point that you have raised. Making it clear that the flat CWND is due to the send buffer size and not the receiver buffer size. We have updated the manuscript accordingly and uploaded a new version.

---

> > > > > > > > > ### Comment · Reviewer_ypqM · 2022-02-11
> > > > > > > > > **Re: Fixed the writing of 5.7**
> > > > > > > > >
> > > > > > > > > This version looks good to me. Approve!

---

> > > > > > > > > > ### Comment · Area_Chair_hHWk · 2022-02-11
> > > > > > > > > > **Congrats!**
> > > > > > > > > >
> > > > > > > > > > It's great that the manuscript is finally approved. Congratulations!
> > > > > > > > > >
> > > > > > > > > > Will contact the program chairs.

---

### Meta-Review · Area_Chair_hHWk · 2021-06-18

**Recommendation:** Revise
**Confidence:** 4

**Metareview:**

Thanks for submitting ALCC to JSYs. Enabling network innovations in application is an important topic. In this context, ALCC makes a decent contribution. That said, reviewers feel strongly that all of the following revision items must be properly addressed in the resubmission. It is CRUCIAL that the authors read the detailed comments of each reviewer for specific requirements of each revision item.

New experiments
- (ypqM, CnU1) Add a baseline that uses a kernel module to enforce a very high CWND (disabling congestion control) only for ALCC traffic and forwards ACKs to the emulated CCs at the application layer. This should be doable, especially as ALCC already uses a kernel module to forward ACKs. This way the CWND will always be higher than the emulated CC, without needing bufferbloat or a loss-driven underlying CC like Cubic.
If there are valid reasons this baseline cannot be implemented (even on Linux), please explain in writing.
Otherwise, please implement it in Linux and experimentally compare it with ALCC (regardless it's better or worse): If ALCC emulates CC performance more closely to the actual one than this baseline does, please show experiment results and explain why. Otherwise, please explain why ALCC is still a better design choice, despite being a less accurate emulation.
- (3y66, 8yrc,CnU1) It is important to show the limitation of ALCC (see writing revisions below). Please show with experiment graphs that ALCC doesn’t work in all cases (i.e., the emulated cellular CCs have much worse performance than their actual performance), when there is no bufferbloat (e.g., random packet loss is higher than 1%). In particular, if ALCC-Verus can ever have worse performance than Verus, please show a graph, otherwise, please rigorously prove that ALCC-Verus is ALWAYS better than native Verus. Please also do the same analysis with respect to Copa and Sprout.
- (3y66, CnU1) As an important sanity check, please plot how the underlying TCP CWND varies over time along with each cellular CC scheme's CWND (if not available, plot sending rate). Use real traces to show how often ALCC receives blocking signals from the underlying TCP and how often Cubic's CWND is higher than each CC's CWND. The expectation is that these incidents are quite rare.
- (3y66) Test ALCC on a real cellular network (NOT Mahimahi-emulated real traces). This should be doable with minor modifications to the current ALCC implementation.  It's important to base the core idea not only on emulation. Report the performance (with repeated runs and error bars) of various CCs over ALCC vs. their native UDP-based performance. The expectation is: ALCC's idea is empirically feasible on most of the tested real networks (ideally from different operators), although it may have glitches on rare occasions (showing real-world glitches is preferred but not required).

Writing revisions
- (8yrc, 3y66, CnU1) Add a section/subsection to clarify the limitations and clearly delineate when CCs built on this framework would perform well and be faithful to an in-datapath implementation (and when they would not). (Additional experiments are needed to show when ALCC doesn't work. See above)
- (ypqM,3y66) 1) Summarize ACK and retransmission mechanisms of different cellular CCs (e.g., Verus and Copa) in detail, then 2) describe how to support these customized ACK mechanisms in existing TCP/IP stack, and finally 3) introduce netfilter based implementation, and also present how ALCC could correctly avoid packet losses / retransmissions.
- (8yrc) Describe the mobile app solution (sec 4.3) and explain how the CC feedback loop is implemented on top of HTTP.

Finally, this is just a minimum revision list. The authors are strongly recommended to make detailed writing clarifications or add experiments to address the rest of the review comments, though this is not required in the revised submission.

---

### Decision · Program_Chairs · 2021-06-16

Revise